# AAK1 activation-mediated iron trafficking drives ferroptotic cell death

Li-Chao Li[1,5], Zhi-Peng Ye[2,5], Ying Xiao[1,3,5], Xian-Ying Zhu[1,3,5], Qia-Qia Li[1], Yi-Qing Guo[1], Huai-Liang Wu[1], Zhi-Ling Li[1], Lin-Yu Wu[1], Yu-Hong Chen[1], Gong-Kan Feng[1], Dong Yang[1], Shan Liu[4], Bing-Xin Hu[1], Jia-Hong Tang[1], Yu-Feng Zhou[1], Jing Li[1], Rong Deng[1] ✉, Hai-Liang Zhang[1] ✉ & Xiao-Feng Zhu[1] ✉

Ferrous iron is necessary for the occurrence of ferroptosis. The molecular mechanisms that maintain iron homeostasis within cells play a crucial role in the regulation of ferroptosis. However, how cells regulate iron uptake during ferroptosis remains unclear. Here, PKCβII is identified as a key kinase mediating transferrin receptor 1 (TFR1) endocytosis through phosphorylation and activation of AP2-associated protein kinase 1 (AAK1) during the ferroptotic process. Mechanistically, activated AAK1 phosphorylates AP2M1, which facilitates the recruitment of clathrin to mediate the endocytosis of TFR1, increasing the levels of both cellular total iron and ferrous iron and thereby promoting ferroptosis. The non-phosphorylatable mutation of AAK1 inhibits ferroptosis and consequently promotes breast tumor growth in vivo. In conclusion, we identify that the PKCβII-AAK1-AP2M1 pathway is a crucial mechanism for the regulation of cellular iron uptake during ferroptosis, which is correlated with the prognosis of breast cancer patients and presents a potential target for cancer therapy.

Ferroptosis is a form of regulated cell death distinct from apoptosis, mediated by iron-dependent lipid peroxidation of lethal levels[1–3]. Increasing evidences implicate potential functions of ferroptosis in multiple pathological conditions, including tumors, ischemia-reperfusion injury, and neurodegenerative diseases, which present distinct potential targets for clinical therapy[4–8]. Previous studies indicated that the accumulation of lipid peroxides resulting from a predominance of production over clearance is essential for the initiation of ferroptosis[9–13]. Polyunsaturated fatty acid (PUFA)-containing PLs (PUFA-PLs) are the most important substrates for the generation of lipid peroxides, which are necessary to execute ferroptosis[14–16]. Enzymes associated with the synthesis of PUFA-PLs, such as ACSL4 and LPCAT3, have been shown to play critical roles in driving ferroptosis[17–19]. On the other hand, tumor cells confer resistance against ferroptosis through surveillance mechanisms for clearance of lipid peroxides. GPX4 reduces lipid peroxides to phospholipid ethanol with glutathione, whose synthesis depends on the uptake of extracellular cystine through SLC7A11, thereby inhibiting ferroptosis[20–22]. Free radical scavenging antioxidants produced by FSP1, DHODH, and GCH1 also contribute to inhibition of ferroptosis[23–26].

In addition to the accumulation of lipid peroxides, ferrous iron is another essential requirement for ferroptosis, which is primarily based on its catalysis on the Fenton reaction and roles as a cofactor of key enzymes involved in lipid peroxidation, such as ALOX, NOX, and

[1]State Key Laboratory of Oncology in South China, Guangdong Provincial Clinical Research Center for Cancer, Guangdong Key Laboratory of Nasopharyngeal Carcinoma Diagnosis and Therapy, Sun Yat-sen University Cancer Center, Guangzhou, China. [2]Hepatology Unit, Departments of Infectious Disease, Guangzhou Women and Children's Medical Center, Guangzhou Medical University, Guangzhou, China. [3]Department of Intensive Care Unit, Sun Yat-sen University Cancer Center, Guangzhou, China. [4]Department of Lymphoma, Guangdong Academy of Medical Sciences, Guangdong Provincial People's Hospital, Guangzhou, China. [5]These authors contributed equally: Li-Chao Li, Zhi-Peng Ye, Ying Xiao, Xian-Ying Zhu. ✉e-mail: dengrong@sysucc.org.cn; zhanghl@sysucc.org.cn; zhuxfeng@mail.sysu.edu.cn

POR[10,27,28]. Under normal conditions, cellular iron is predominantly stored in the form of $Fe^{3+}$ complexed with ferritin, serving as an iron reservoir. Mechanisms that regulate ferritin utilization influence the levels of $Fe^{2+}$ in cells, thereby affecting sensitivity to ferroptosis[27–29]. Generally, enhanced degradation of ferritin leads to increased intracellular divalent iron levels, whereas increased synthesis of ferritin produces the opposite effect, which causes higher or lower vulnerability to ferroptosis respectively[30–34]. However, the induction of ferroptosis in cancer cells typically leads to increased ferritin levels, which paradoxically contradicts the anticipated rise in ferritin degradation[31,35,36]. It suggests that extracellular iron uptake may serve as a critical iron supply for the initiation of ferroptosis. Given the essential role of iron in Fenton reaction, which mediates lipid peroxidation, it is significant to elucidate the iron source in the induction of ferroptosis.

Transferrin receptor 1 (TFR1) is a homo-dimeric glycoprotein receptor composed of two subunits, each capable of binding one $Fe^{3+}$-loaded transferrin[27]. Clathrin-mediated endocytosis of the TFR1-transferrin complex is the primary mechanism through which tumor cells absorb iron from the extracellular environment. V-ATPase acidifies the endosomes formed by TFR1 endocytosis and facilitates the release of iron from transferrin, thus providing a source for iron utilization or storage[29]. Previous studies have shown that *RAS* enhances cellular $Fe^{2+}$ levels by upregulating *TFR1* expression, thereby increasing the sensitivity of tumor cells to ferroptosis[37]; conversely, the knockout of *TFR1* significantly inhibits ferroptosis induced by erastin or cysteine deprivation[38]. Additionally, a large-scale antibody screening study identified TFR1 antibody (3F3-FMA) as a potential specific biomarker of ferroptosis[39]. In summary, TFR1 is a key molecule for iron uptake in tumor cells whose function on ferroptosis has been definitely implicated. However, the specific mechanisms governing TFR1 endocytosis and iron uptake during ferroptosis remain unclear.

AP2-associated protein kinase 1 (AAK1) is a kind of Ser/Thr kinase that belongs to the family of adaptor proteins. It was reported to promote clathrin-mediated endocytosis through the phosphorylation of AP2M1 which enhances its binding affinity to cargo membrane proteins[40–43]. In this study, we found that PKCβII facilitates clathrin-mediated endocytosis of TFR1 and increases cellular iron levels through the phosphorylation of AAK1, thus promoting ferroptosis of tumor cells. Combined with our previous findings identifying PKCβII functions as a sensor of lipid peroxidation[44], this study demonstrates its dual role in ferroptosis regulation - not only detecting lipid peroxides but also actively coordinating cellular iron acquisition. These findings establish a mechanistic link between lipid peroxidation and iron dyshomeostasis, revealing their synergistic interplay in driving ferroptosis execution.

## Results

### PKCβII promotes the endocytosis of TFR1 and iron uptake in the ferroptotic process

Intracellular labile iron pools exhibit a significant elevation during the induction of ferroptosis[27–29]. However, we observed that the levels of ferritin were generally increased during ferroptosis as previously reported, which paradoxically contradicts the expected rise in ferritin degradation[31,32,36] (Fig. 1A). Consistent with previous research findings, both cellular total iron and ferrous iron levels increased in the ferroptotic process[27–29] (Fig. 1B, C). This result prompted us to explore the exogenous pathway of iron supply during the induction of ferroptosis. As a primary pathway for extracellular iron uptake, the regulation of TFR1 endocytosis was poorly understood. To detect the endocytosis of TFR1, we performed endocytosis detection methods previously reported[45]. We labelled surface TFR1 with TFR1-specific antibodies at 4 °C followed by transferring cells to 37 °C to allow TFR1 internalization. The internalization level of TFR1 will be established by antibody-labelled TFR1 from the cell surface, evaluated by flow cytometry

(Fig. 1D). To explore the universal significance of endocytosis of TFR1 in the regulation of ferroptosis, we analyzed the endocytosis of TFR1 during erastin-induced ferroptosis in a panel of cancer cell lines. The result revealed that the endocytosis of TFR1 was commonly elevated during ferroptosis and reserved by dynasore, an endocytosis inhibitor which competitively blocks the GTPase activity of dynamin[46,47] (Fig. 1E). Together, these findings suggest that enhanced endocytosis of TFR1, which mediates iron uptake, represents a hallmark of ferroptosis.

Our previous work has identified PKCβII as a critical sensor of lipid peroxidation in ferroptosis, capable of facilitating the accumulation of lipid peroxides through the phosphorylation of ACSL4 and thereby promoting ferroptosis[44]. In the same time, we observed that both cellular total iron and divalent iron showed a lower level in *PKCβ*-knockout MDA-MB-231 and HT1080 cells treated with erastin (Fig. 1F, G and Supplementary Fig. 1A). Decreased ferrous iron was also verified by immunofluorescence using $Fe^{2+}$-sensitive probe in *PKCβ*-knockout cells (Supplementary Fig. 1B, C). Based on these results, we hypothesize that PKCβ may regulate iron levels through the effect on the endocytosis of TFR1. Under resting-state condition, we found that knockout of *PKCβ* did not affect the levels of surface TFR1 and its endocytosis (Supplementary Fig. 1D, E). However, *PKCβ*-knockout cells exhibited higher levels of surface TFR1 during erastin-induced ferroptosis, indicating the suppression of its endocytosis (Fig. 1H, I). Remarkable inhibition of TFR1 endocytosis was also observed in *PKCβ*-knockout cells in incubation-time-course analysis (Supplementary Fig. 1H). In addition, although the endocytosis of TFR1 was facilitated in a concentration- and time-dependent manner during ferroptosis induced by erastin, it was significantly inhibited in *PKCβ*-knockout cells. (Supplementary Fig. 1I, J). Moreover, synergistic effect of blocking endocytosis and depleting *PKCβ* was not observed (Supplementary Fig. 1F). Additionally, incubation with primaquine, an inhibitor of endocytic recycling[45], induced an extra reduction of surface TFR1, suggesting that a proportion of internalized TFR1 was recycling (Supplementary Fig. 1G). These results indicated that the endocytosis of TFR1 was enhanced during ferroptosis in a PKCβ-dependent manner. *PKCβ* has two transcripts, *PKCβI* and *PKCβII*. The latter includes a 50-amino-acid hydrophobic sequence at its C-terminus, which may contribute to its enhanced activation and improved membrane localization during ferroptosis[44,48]. Therefore, we tested which transcript regulated the endocytosis of TFR1 by rescuing *PKCβ*-knockout cells with single guide RNA (sgRNA)-resistant *PKCβI* or *PKCβII*. Notably, *PKCβII*, but not *PKCβI*, almost completely reversed the inhibitory effect of *PKCβ* knockout on the endocytosis of TFR1 (Fig. 1J, K). To further confirm the function of PKCβII on the endocytosis of TFR1, we also found that the rescuing effect of PKCβII on the endocytosis of TFR1 in *PKCβ*-knockout cells was reversed by dynasore, an endocytosis inhibitor which competitively blocks the GTPase activity of dynamin[46,47] (Supplementary Fig. 1K). Expectedly, *PKCβII* not *PKCβI*, reversed the inhibitory effect of *PKCβ* knockout on iron levels (Fig. 1L and Supplementary Fig. 1L). These results indicated that the endocytosis of TFR1 linked to ferroptosis was enhanced by PKCβII. Taken together, these findings suggest that PKCβII promotes the endocytosis of TFR1 linked to ferroptosis, thereby increasing intracellular iron levels.

### PKCβII promotes the endocytosis of TFR1 and iron uptake through the interaction with AAK1

To explore the mechanism by which *PKCβ* regulate the endocytosis of TFR1, we observed the phosphorylation levels of proteins in *PKCβ*-knockout cells through phosphoproteomic analysis (Fig. 2A). ACSL4, the previously identified downstream protein of PKCβII, was also detected in the proteomic analysis, confirming the validation of the results. In addition, we found that AAK1, an endocytosis-related kinase, exhibited reduced phosphorylation levels in *PKCβ*-knockout cells. AAK1 belongs to the family of adaptor proteins with Ser/Thr kinase activity. It has been indicated that AAK1 catalyzes the phosphorylation

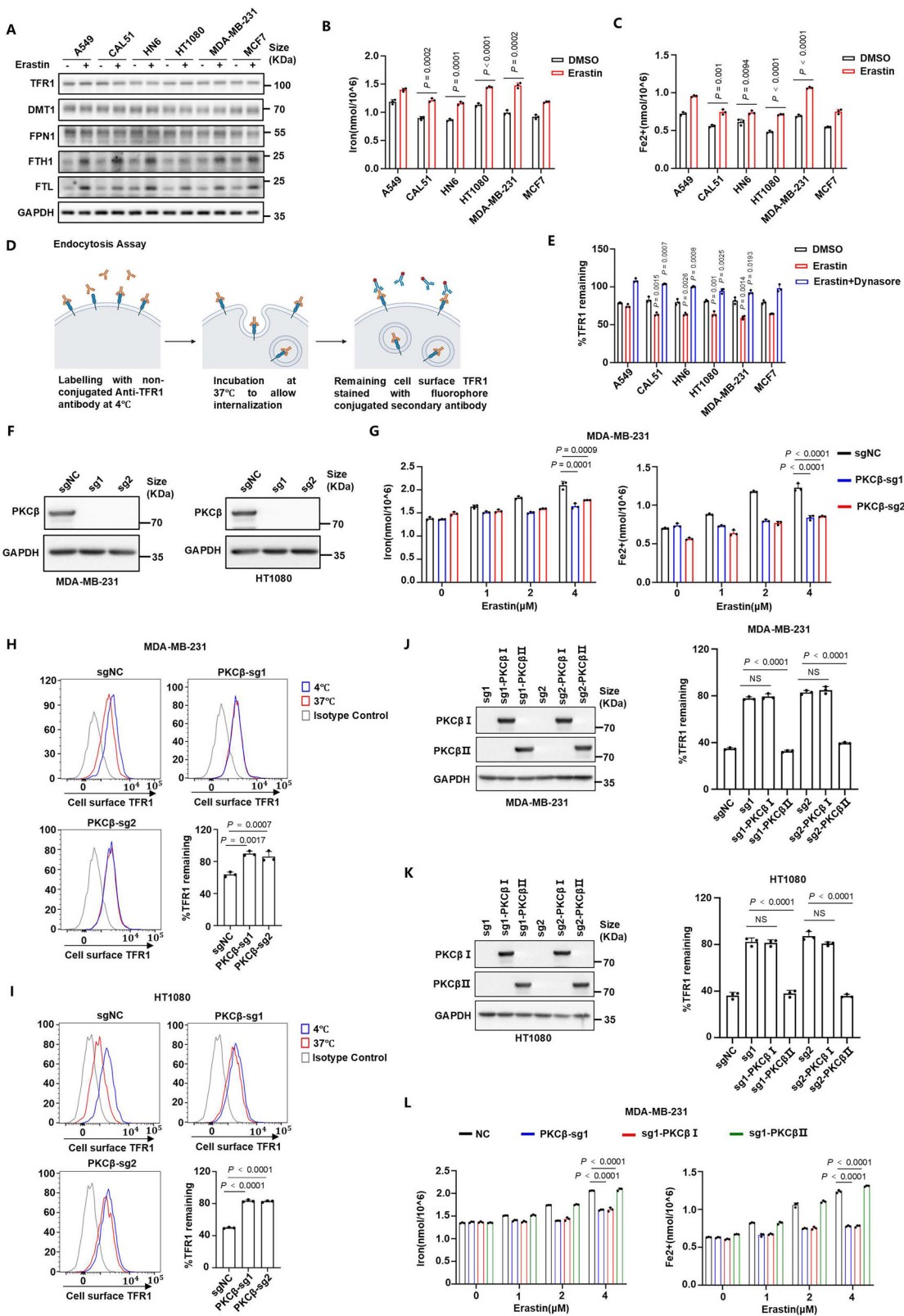

of AP2M1 to enhance its binding affinity to cargo membrane proteins, thereby triggering clathrin assembly and facilitating the process of endocytosis[40–43]. Therefore, we next examined its effect on the endocytosis of TFR1. Under resting-state condition, we found that knockout of *AAK1* did not affect the levels of surface TFR1 or its endocytosis in MDA-MB-231 and HT1080 cells (Fig. 2B and Supplementary Fig. 2A, B). However, *AAK1*-knockout cells exhibited higher levels of surface TFR1

during erastin-induced ferroptosis, indicating that the endocytosis of TFR1 was inhibited (Fig. 2C, D). Incubation-time-course analysis also revealed significant inhibition of endocytosis of TFR1 in *AAK1*-knockout cells (Supplementary Fig. 2D). Additionally, although the endocytosis of TFR1 was facilitated in a concentration- and time-dependent manner during erastin-induced ferroptosis, it was remarkably inhibited in *AAK1*-knockout cells (Supplementary Fig. 2E, F). Moreover,

**Fig. 1 | PKCβII promotes the endocytosis of TFR1 and iron uptake in the ferroptotic process. A** Protein levels associated with iron metabolism in various cancer cell lines treated with erastin for 12 h. A549, 8 μM; CAL51, 30 μM; HN6, 30 μM; HT1080, 4 μM; MDA-MB-231, 8 μM; MCF7, 30 μM. **B, C** Total (**B**) and divalent (**C**) iron levels of various cancer cell lines treated with erastin for 12 h. A549, 4 μM; CAL51, 20 μM; HN6, 20 μM; HT1080, 1 μM; MDA-MB-231, 2 μM; MCF7, 20 μM. **D** Schematic of the endocytosis assays. TFR1 in the cell surface were labelled with TFR1-specific antibodies at 4 °C followed by transferring cells to 37 °C to allow TFR1 internalization. The internalization levels of TFR1 will be established by antibody-labelled TFR1 from the cell surface evaluated by flow cytometry. This figure was created in BioRender. Lichao, L. (2025) https://BioRender.com/p1ntlvc.
**E** Endocytosis assays of TFR1 in various cancer cell lines treated with erastin for 12 h. A549, 4 μM; CAL51, 20 μM; HN6, 20 μM; HT1080, 1 μM; MDA-MB-231, 2 μM; MCF7, 20 μM. **F** *PKCβ* was knocked out using single guide RNAs (sgRNAs) in MDA-MB-231 (left) and HT1080 (right) cells. **G** Total (left) and divalent (right) iron levels in the indicated MDA-MB-231 cells treated with erastin for 12 h. **H, I** Endocytosis assays of TFR1 in the indicated MDA-MB-231 (**H**) and HT1080 (**I**) cells treated with 2 μM or 1 μM erastin for 12 h, respectively. The cells with blue highlight were kept in 4 °C for dormancy. The cells with red highlight were transferred to 37 °C for internalization. The cells with grey highlight were labelled with non-specific antibody as isotype control. **J, K** Plasmids of *PKCβI* or *PKCβII* were transfected into *PKCβ*-knockout MDA-MB-231 (**J**) and HT1080 (**K**) cells, respectively. Endocytosis assays of TFR1 were performed in these cells treated with 2 μM or 1 μM erastin for 12 h, respectively. **L** Total (left) and divalent (right) iron levels in the indicated MDA-MB-231 cells treated with erastin for 12 h. **A, F, J, K** Data are representative of *n* = 3 biologically independent experiments. **B, C, E, J, K** Data are presented as means ± SD, *n* = 3 biologically independent experiments, unpaired two-tailed Student's *t* test. **G–I, L** Data are presented as means ± SD, *n* = 3 biologically independent experiments, one-way ANOVA test.

synergistic effect of blocking endocytosis and depleting *AAK1* was not observed (Supplementary Fig. 2C). To further confirm the role of AAK1 on the endocytosis of TFR1, we performed endocytosis assays in *AAK1*-knockout cells transfected with sgRNA-resistant *AAK1* (Fig. 2E, F). Subsequently, the re-expression of *AAK1* rescued the endocytosis of TFR1 suppressed by knockout of *AAK1*, but was reversed by treatment with dynasore (Fig. 2E, F). These results indicated that the endocytosis of TFR1 linked to ferroptosis was enhanced by AAK1. Furthermore, we assessed the impact of AAK1 on intracellular iron levels during ferroptosis. Consistent with the results of PKCβII, knockout of *AAK1* inhibited the levels of total and ferrous iron during ferroptosis induced by erastin, while the re-expression of *AAK1* reversed the iron reductions. (Fig. 2G and Supplementary Fig. 3A–D). These results were also confirmed by immunofluorescence using the $Fe^{2+}$-sensitive probe (Supplementary Fig. 3E, F). Taken together, these findings suggest that AAK1 promotes the endocytosis of TFR1 and increases cellular iron levels during ferroptosis.

As the phosphoproteomic analysis showed that AAK1 was a potential downstream molecule of PKCβII, we next investigated whether there was a direct interaction between PKCβII and AAK1. We performed co-immunoprecipitation experiments through pulling down endogenous PKCβII or AAK1 in MDA-MB-231 and HT1080 cells. Increased binding of PKCβII with both endogenous or exogenous AAK1 was detected in a concentration-dependent manner during erastin-induced ferroptosis (Fig. 2H and Supplementary Fig. 4A). Similarly, co-immunoprecipitation by pulling down endogenous AAK1 also demonstrated increased binding of AAK1 with endogenous or exogenous PKCβII during ferroptosis induced by erastin in a concentration-dependent manner (Fig. 2I and Supplementary Fig. 4B). However, this binding was substantially decreased with the treatment of two PKC inhibitors, Go6983 or enzastaurin (Supplementary Fig. 4C, D), suggesting that the formation of the PKCβII–AAK1 complex was dependent on the activation of PKCβII[49,50]. Immunofluorescent staining for PKCβII and AAK1 using specific antibodies also revealed increased colocalization during erastin-induced ferroptosis in a concentration- and time-dependent manner (Supplementary Fig. 4G–J). Interestingly, we also observed a weak interaction between PKCβI and AAK1, which was not increased during ferroptosis, suggesting that PKCβI may not be involved in the ferroptotic process (Supplementary Fig. 4E, F). Taken together, PKCβII promotes the endocytosis of TFR1 and iron uptake through the interaction with AAK1.

### Phosphorylation of S670/T674 by PKCβII is critical for the activation of AAK1

PKCβII was identified to play a key role driving ferroptosis. Given the results that PKCβII interacted with and phosphorylated AAK1, we aimed to explore the phosphorylation levels of AAK1 during ferroptosis. We observed that AAK1 exhibited increased total phosphorylation levels during ferroptosis induced by erastin in a concentration-

and time-dependent manner in MDA-MB-231 and HT1080 cells, which was inhibited by two PKC inhibitors Go6983, enzastaurin, or genetic knockout of *PKCβ* (Fig. 3A–C and Supplementary Fig. 5A–C). As expected, only PKCβII, not PKCβI, was able to restore the phosphorylation of AAK1 (Fig. 3D and Supplementary Fig. 5D). These results indicated that the elevated phosphorylation levels of AAK1 linked to ferroptosis was dependent on PKCβII. Moreover, the phosphorylation of AAK1 was suppressed under the condition that ferroptosis was blocked by Fer-1 (Supplementary Fig. 5E). To provide direct evidence of the interaction of PKCβII and AAK1, we performed in vitro kinase assays to confirm whether PKCβII directly phosphorylated AAK1. The level of Ser/Thr phosphorylation of AAK1 protein increased significantly in the presence of recombinant active PKCβII kinase, but was blocked in the addition of λ-phosphatase (Fig. 3E, F). In summary, PKCβII directly interacts with and phosphorylates AAK1 during ferroptosis. The total phosphorylation levels of AAK1 were increased during ferroptosis, which was dependent on PKCβII.

The phosphoproteomic analysis suggested that S670/T674 may serve as phosphorylation sites of AAK1. To determine the specific site at which AAK1 is phosphorylated by PKCβII, we generated an AAK1 exogenous expression cell model in which potential phosphorylation sites S670/T674 on AAK1 were both mutated to non-phosphorylatable Ala (phosphorylation inactivation mutation, Ser/Thr to Ala) or Glu (phosphorylation activation mutation, Ser/Thr to Glu), or mutated alone (Fig. 3G–I). We next examined the impact of mutant AAK1 on the endocytosis of TFR1. In *PKCβ*- or *AAK1*-knockout cells transfected with various mutant *AAK1*, only the S670/T674E mutation restored the endocytosis of TFR1 during erastin-induced ferroptosis (Fig. 3J and Supplementary Fig. 5F). These results indicated that the role of AAK1 in the endocytosis of TFR1 linked to ferroptosis was dependent on the phosphorylation of S670/T674 dual sites by PKCβII. To further detect endogenous phosphorylation of AAK1 at S670/T674, a phospho-S670/T674-AAK1-specific (p-AAK1 (S670/T674)) antibody was generated and validated (Supplementary Fig. 5G). The phosphorylation of AAK1 was detected using antibody against p-AAK1(S670/T674) in the in vitro kinase assay system added with wild-type AAK1 protein but not S670/T674Ala mutant AAK1 (Fig. 3K). These results revealed that PKCβII directly phosphorylated AAK1 at S670/T674 dual sites. In addition, the elevated phosphorylation of AAK1 at S670/T674 dual sites during erastin-induced ferroptosis was inhibited by Go6983, enzastaurin, or Fer-1, suggesting that the phosphorylation of AAK1 at S670/T674 sites results from ferroptosis mediated by PKCβII (Fig. 3L, M). Together, these results suggest that phosphorylation of AAK1 at S670/T674 sites is critical for the endocytosis of TFR1 and ferroptosis in a PKCβII dependent manner.

### PKCβII-AAK1-AP2M1 pathway is essential for the endocytosis of TFR1 and iron uptake during ferroptosis

Given that AAK1-mediated phosphorylation of AP2M1 at Thr156 site enhances the interaction of clathrin and cargo membrane proteins to

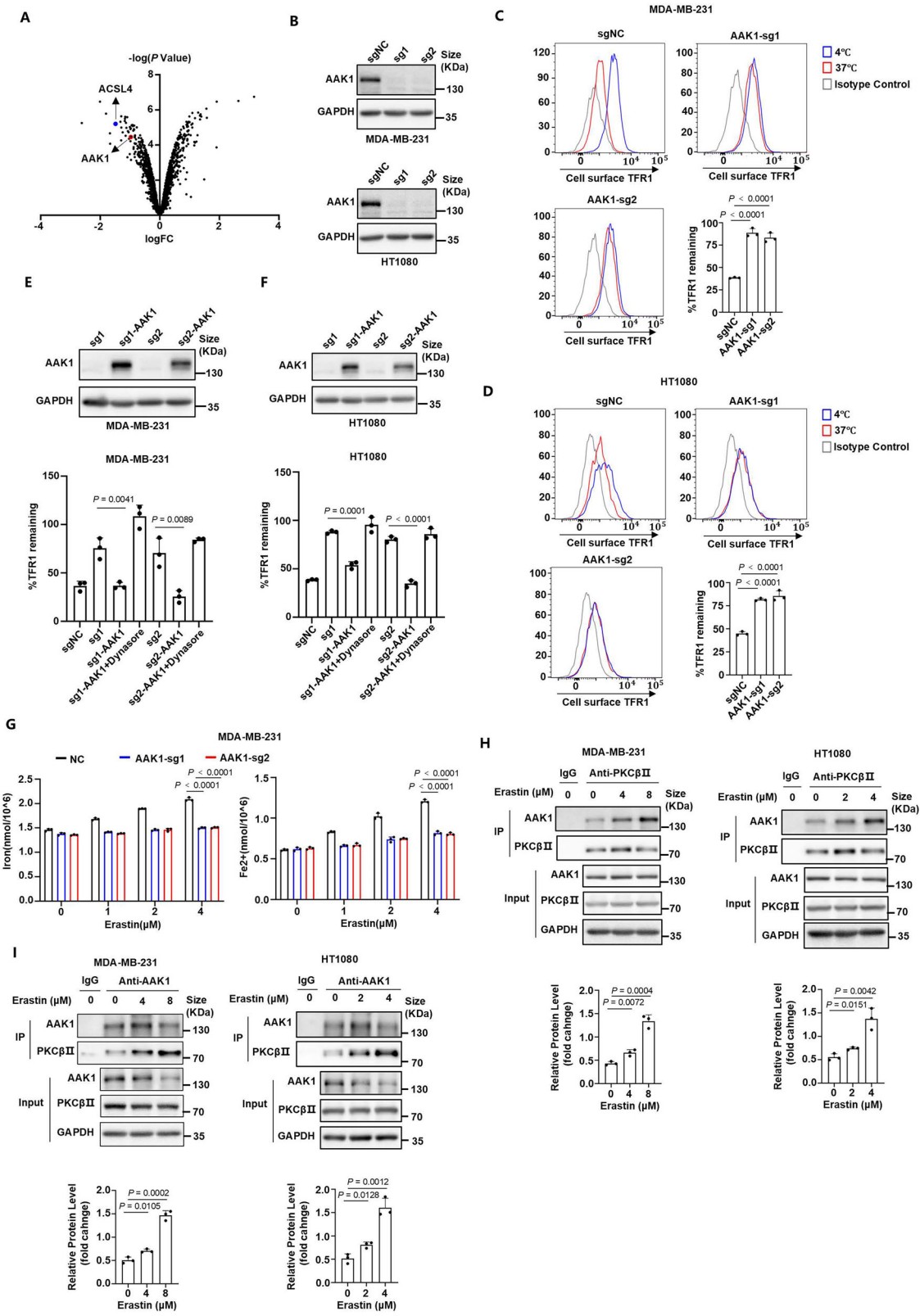

drive endocytosis[40–43], we aimed to confirm that the PKCβII-AAK1-AP2M1 pathway is crucial for the endocytosis of TFR1 in the induction of ferroptosis. We found that the phosphorylation of AP2M1 was significantly increased during ferroptosis, which was inhibited by knockout of *PKCβ* or treatment with Go6983 (Fig. 4A, B). Consistent with pervious results, only PKCβII but not PKCβI restored the phosphorylation of AP2M1 in *PKCβ*-knockout cells during ferroptosis

(Fig. 4C). Furthermore, the phosphorylation level of AP2M1 was also inhibited by knockout of *AAK1* or treatment with AAK1 inhibitor, SGC-AAK1-1 during ferroptosis[40,51] (Fig. 4D, E). As expected, only the wild-type or S670/T674E mutant AAK1 restored the phosphorylation of AP2M1 in *AAK1*-knockout cells during ferroptosis (Fig. 4F). These results indicated that the phosphorylation of AP2M1 was increased during ferroptosis through PKCβII-AAK1 axis.

**Fig. 2 | PKCβII promotes the endocytosis of TFR1 and iron uptake through the interaction with AAK1. A** Phosphorylation level and gene score for individual genes were analyzed in the phosphoproteome using three individual wild-type and *PKCβ*-KO MDA-MB-231 cells. Potential genes exhibiting lower phosphorylation level have a negative score and genes exhibiting higher phosphorylation level have a positive score. Red highlights *AAK1*. Blue highlights *ACSL4*. Data was analyzed by unpaired two-tailed Student's *t* test. **B** AAK1 was knocked out by single guide RNAs (sgRNAs) in MDA-MB-231 (top) and HT1080 (bottom) cells. **C, D** Endocytosis of TFR1 in the indicated MDA-MB-231 (**C**) and HT1080 (**D**) cells treated with 2 μM or 1 μM erastin for 12 h, respectively. The cells with blue highlight were kept in 4 °C for dormancy. The cells with red highlight were transferred to 37 °C for internalization. The cells with grey highlight were labelled with non-specific antibody as isotype control. **E** *AAK1*-knockout MDA-MB-231 cells were transfected with sgRNA-resistant *AAK1*. Endocytosis of TFR1 were performed in the indicated cells treated with erastin with/without dynasore. erastin, 2 μM for 12 h; dynasore, 150 μM for 1 h. **F** *AAK1*-knockout HT1080 cells were transfected with sgRNA-resistant *AAK1*. Endocytosis of TFR1 were performed in the indicated cells treated with erastin with/without dynasore. erastin, 1 μM for 12 h; dynasore, 150 μM for 1 h. **G** Total (left) and divalent (right) iron levels were assayed in the indicated MDA-MB-231 cells treated with erastin for 12 h. **H** Endogenous PKCβII was immunoprecipitated from MDA-MB-231 (left) and HT1080 (right) cells treated with erastin for 12 h, followed by immunoblots using a AAK1-specific antibody to establish the interaction of endogenous PKCβII with endogenous AAK1. **I** Endogenous AAK1 was immunoprecipitated from MDA-MB-231 (left) and HT1080 (right) cells treated with erastin for 12 h, followed by immunoblots using a PKCβII-specific antibody to establish the interaction of endogenous AAK1 with endogenous PKCβII. **B–F, H, I** Data are representative of *n* = 3 biologically independent experiments. **C, D, G** Data are presented as means ± SD, *n* = 3 biologically independent experiments, one-way ANOVA test. **E, F, H, I** Data are presented as means ± SD, *n* = 3 biologically independent experiments, unpaired two-tailed Student's *t* test.

To further explore effect of AP2M1 on the endocytosis of TFR1, *AP2M1*-knockout cells were generated and transfected with wild-type or T156 mutant AP2M1 in MDA-MB-231 and HT1080 cells (Fig. 4G and Supplementary Fig. 6A). As expected, knockout of *AP2M1* inhibited the endocytosis of TFR1 during erastin-induced ferroptosis, which could be restored only by wild-type or T156E mutant AP2M1 (Fig. 4H and Supplementary Fig. 6B). Additionally, over-expression of T156E mutant AP2M1 restored the endocytosis of TFR1 in *PKCβ*- or *AAK1*- knockout cells during ferroptosis. (Supplementary Fig. 6C, D). However, over-expression of wild-type AP2M1 also resulted in a partial recovery of TFR1 endocytosis in *PKCβ*-knockout cells, possibly due to the involvement of other kinases related to endocytosis. These results indicated that the cascade phosphorylation of PKCβII-AAK1-AP2M1 axis is vital for the endocytosis of TFR1 and iron uptake during ferroptosis. In summary, PKCβII-AAK1-AP2M1 pathway functions as a critical regulatory mechanism for iron supply during ferroptosis.

## PKCβII-AAK1-AP2M1 pathway significantly enhances the sensitivity to ferroptosis

Considering that AAK1 serves as a key role in iron uptake, we then aim to validate AAK1 as a contributor of ferroptosis. Firstly, we found that genetic knockdown of *AAK1* by specific siRNA in MDA-MB-231 and HT1080 cells significantly led to the inhibition of erastin- or cystine deprivation- induced ferroptosis and lipid peroxidation (Fig. 5A–D and Supplementary Fig. 7A, B). Additionally, knockout of *AAK1* also inhibited ferroptosis and lipid peroxidation (Fig. 5E, F and Supplementary Fig. 7C, D). Subsequently, *AAK1*-rescued cells were treated with erastin or cystine deprivation alone or in combination with inhibitors of various kinds of cell death. As expected, erastin- or cystine deprivation-induced lipid peroxidation and cell death could be fully rescued by the ferroptosis inhibitor ferrostatin-1 (Fer-1; lipid peroxidation scavenger), the iron chelator deferoxamine (DFO), or the antioxidant *N*-acetylcysteine but not by the inhibitors of apoptosis or necroptosis (Fig. 5G, H and Supplementary Fig. 7E, F). In addition, SGC-AAK1-1 also inhibited lipid peroxidation and cell death induced by erastin or cystine deprivation (Supplementary Fig. 8A, B). These results indicated that AAK1 is a critical contributor of ferroptosis. We further detected whether the effect of AAK1 on ferroptosis depends on phosphorylation at S670/T674 sites. In *AAK1*-knockout cells re-expressing wild-type or various mutant *AAK1*, we found that only wild-type or S670/T674E mutant *AAK1* restored sensitivity to ferroptosis and lipid peroxidation compared to other mutations (Fig. 5I, J and Supplementary Fig. 8C, D). Together, these results indicate that the phosphorylated S670/T674 sites of AAK1 by PKCβII is critical for the induction of ferroptosis.

To further demonstrate the effect of AP2M1 on ferroptosis, we examined the lipid peroxidation and cell death induced by erastin or cystine deprivation in *AP2M1*-knockout and reconstituted cells. Expectedly, knockout of *AP2M1* inhibited ferroptosis and lipid peroxidation, which was restored only by reconstitution with wild-type or T156E mutant *AP2M1* (Fig. 5K, L and Supplementary Fig. 8E, F). These results revealed that AP2M1 also had a positive impact on ferroptosis. Taken together, PKCβII-AAK1-AP2M1 pathway significantly enhances the sensitivity to ferroptosis.

## Activation of PKCβII-AAK1-AP2M1 pathway inhibits tumor growth through the induction of ferroptosis in vivo

To investigate the clinical significance of PKCβII-AAK1-AP2M1 pathway, we further evaluated the prognostic value of *AAK1* expression in breast cancer patients. Kaplan–Meier survival analysis indicated that the high expression of PKCβII-AAK1-AP2M1 pathway was significantly associated with prolonged overall survival and relapse-free survival in breast cancer patients (Fig. 6A, B and Supplementary Fig. 9A, B). Moreover, we found that the expression of *AAK1* was lower in primary tumor tissues of breast cancer than normal tissues (Fig. 6C). These results suggest that PKCβII-AAK1-AP2M1 pathway might be a potential tumor prognostic indicator. Additionally, we observed a negative correlation between *AAK1* and gene set containing anti-ferroptosis genes analyzed by TCGA and ICGC-BC dataset (Supplementary Fig. 9C, D). Taken together, these results suggest that the expression of *AAK1* might be a potential tumor prognostic indicator related to the susceptibility of ferroptosis in breast tumors. To further explore the association between PKCβII and the phosphorylation level of AAK1 in tumor tissues, we performed immunohistochemical analysis in 217 human TNBC specimens using anti-PKCβII and anti-p-AAK1 (S670/T674) antibodies. We observed a positive correlation between PKCβII and the phosphorylation level of AAK1 (Fig. 6D, E). In addition, we also found a positive correlation between PKCβII and 4-HNE, a marker of lipid peroxidation and ferroptosis (Fig. 6F). These results reveal the significant roles of PKCβII and AAK1 for the induction of ferroptosis in breast tumors.

To further confirm the role of AAK1 during the process of ferroptosis in vivo, we performed xenograft tumor model by inoculating *AAK1*-knockout MDA-MB-231 cells with or without transfecting plasmids of wild-type or S670/T674Ala mutant *AAK1* into nude mice. These nude mice were treated with IKE (a lipid-soluble form of erastin for animal experiment) or DMSO as control. As expected, knockout of *AAK1* significantly inhibited ferroptosis through resistance of lipid peroxidation induced by IKE and promoted tumor growth (Fig. 6G–J). The inhibitory effect of lipid peroxidation was also demonstrated by reduced levels of *PTGS2* and 4-HNE (Fig. 6K, L). Consistent with the results mentioned above, only wild-type but not S670/T674Ala mutant *AAK1*, restored the sensitivity to ferroptosis of tumors (Fig. 6J–L). In addition, knockout of *AAK1* also remarkably inhibited lipid peroxidation under BSO-mediated ferroptosis and promoted tumor growth in nude mice xenograft model (Supplementary Fig. 9E–J). These results indicated the positive effect of AAK1 on ferroptosis. Taken together,

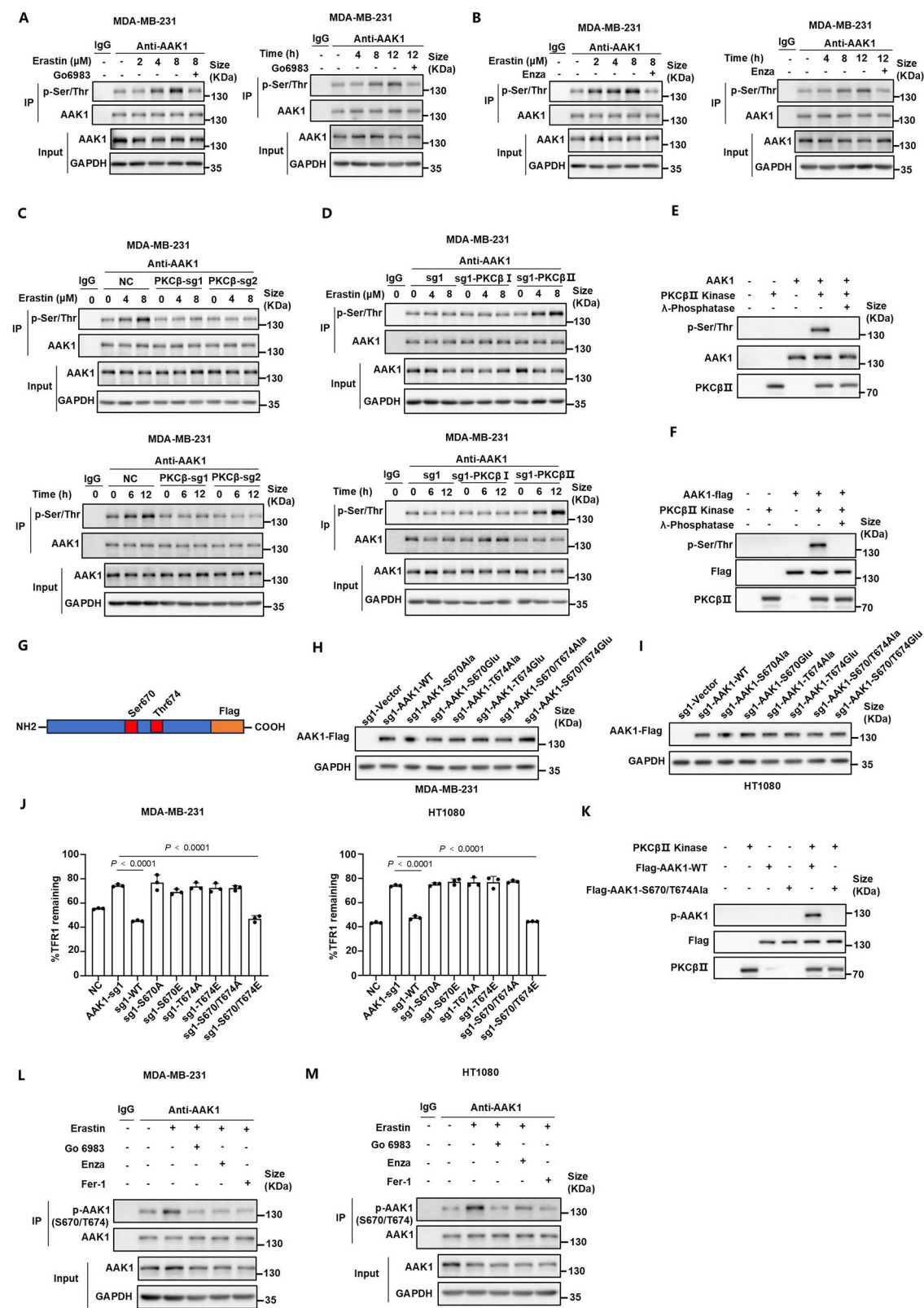

activation of PKCβII-AAK1-AP2M1 pathway inhibits tumor growth through the induction of ferroptosis in vivo.

As ionizing radiation (IR) was reported to induce ferroptosis of tumors in radiotherapy, further experiments were performed to confirm the effect of AAK1 to radiosensitivity in vivo[5,8,10]. *AAK1*-knockout and negative control MDA-MB-231 cells were inoculated into nude mice, followed by treatment with either DMSO or Lipro-1, with or without IR. Expectedly, ferroptosis induced by IR was inhibited by Lipro-1. Knockout of *AAK1* significantly inhibited ferroptosis and lipid peroxidation, thereby promoting tumor growth (Supplementary Fig. 10A−E). These results indicate that AAK1 enhances the radiosensitivity of tumors to ionizing radiation. Activation of PKCβII-AAK1-AP2M1 pathway may produce a synergistic effect combined with radiotherapy.

**Fig. 3 | Phosphorylation of S670/T674 by PKCβII is critical for the activation of AAK1. A** Total phosphorylation levels of AAK1 in MDA-MB-231 cells treated with erastin, with/without 5 μM Go6983. **B** Total phosphorylation levels of AAK1 in MDA-MB-231 cells treated with erastin, with/without 5 μM Enza (enzastaurin). **C, D** Total phosphorylation levels of AAK1 in the indicated MDA-MB-231 cells treated with erastin. **E** Purified recombinant AAK1 proteins were incubated with/without recombinant activated PKCβII kinase or λ-phosphatase for 0.5 h in kinase buffer with ATP in vitro. **F** HEK293T cells were transfected with Flag-AAK1 plasmid, followed by isolating from HEK293T cells using anti-Flag antibody. The isolated AAK1 proteins were incubated with/without recombinant activated PKCβII kinase or λ-phosphatase for 0.5 h in kinase buffer with ATP in vitro. **G** Schematic of the potential phosphorylation sites of AAK1 indicated by phosphoproteomic analysis. **H, I** *AAK1*-knockout MDA-MB-231 (**H**) and HT1080 (**I**) cells were transfected with plasmids of wild-type or various mutant *AAK1*. The phosphorylation inactivated mutation was performed by mutating Ser/Thr into Ala, while the activated mutation was performed by mutating Ser/Thr into Glu. Single or double mutants of both of

the activated or inactivated mutation of potential phosphorylation sites S670/T674 on AAK1 were included. **J** Endocytosis assays of TFR1 in the indicated MDA-MB-231 (left) and HT1080 (right) cells treated with 2 μM or 1 μM erastin, respectively for 12 h. **K** HEK293T cells were transfected with wild-type Flag-AAK1 or mutant Flag-AAK1-S670/T674Ala plasmid, followed by isolating from HEK293T cells using anti-Flag antibody. The isolated AAK1 proteins were incubated with/without recombinant activated PKCβII kinase for 0.5 h in kinase buffer with ATP in vitro. The phosphorylation levels of AAK1 were established by a phospho-S-670/T674-AAK1-specific (p-AAK1 (S670/T674)) antibody. **L, M** The phosphorylation levels of AAK1 in MDA-MB-231 (**L**) and HT1080 (**M**) cells treated with erastin with/without Go6983, Enza, or Fer-1 for 12 h by Immunoblots using phospho-S670/T674-AAK1-specific antibody. erastin, Go6983, Enza, Fer-1, 10 μM for MDA-MB-231 and 5 μM for HT1080. **A–F, H, I, K–M** Data are representative of $n = 3$ biologically independent experiments. **J** Data are presented as means ± SD, $n = 3$ biologically independent experiments, unpaired two-tailed Student's $t$ test.

## Discussion

In this study, we identified enhanced interaction between PKCβII and AAK1 during ferroptosis by phosphoproteomic analysis performed in *PKCβ*-knockout MDA-MB-231 cells. PKCβII directly phosphorylates AAK1 at S670/T674 dual sites, which in turn promotes the phosphorylation of AP2M1 by AAK1, facilitating clathrin-mediated endocytosis of TFR1 and iron uptake (Fig. 7). Ferrous iron is an essential requirement for the initiation of ferroptosis. Increased iron uptake leads to elevated cellular iron levels, ultimately promoting the accumulation of lipid peroxides and ferroptosis[10,27,28]. The ablation of *AAK1* or non-phosphorylatable mutants blocking the PKCβII-AAK1-AP2M1 axis can effectively inhibit ferroptosis in vivo and promote tumor growth.

The accumulation of lipid peroxides, which depends on iron metabolism, is indispensable for the execution of ferroptosis. Iron metabolism plays a crucial role in the regulation of ferroptosis. We previously discovered that AKT inhibits the degradation of TRPML1 through its phosphorylation, promoting lysosomal exocytosis and reducing cellular iron levels, thus mediating tumor resistance to ferroptosis[52]. Although numerous studies have revealed the key role of iron metabolism in ferroptosis, their interests have primarily focused on the utilization and storage of intracellular iron. Generally, increased degradation of ferritin results in elevated intracellular levels of divalent iron, leading to higher vulnerability to ferroptosis, while increased synthesis of ferritin has the opposite effect[30–32]. Unexpectedly, increased ferritin levels are typically observed during ferroptosis[31,35,36], indicating that extracellular iron uptake may serve as a critical iron supply for the initiation of ferroptosis. However, the specific molecular mechanisms regulating iron uptake during ferroptosis by which cells adapt to elevated iron demand for lipid peroxidation, remain poorly understood. In this study, we discovered that cancer cells absorb iron from the extracellular environment through PKCβII phosphorylating AAK1. This finding significantly broadens our understanding of the sources of iron during ferroptosis. Cancer cells primarily acquire iron from the extracellular environment through clathrin-mediated endocytosis of TFR1[29]. We demonstrated that enhanced endocytosis of TFR1 mediated by the PKCβII-AAK1-AP2M1 axis during ferroptosis represents a critical pathway through which cells actively absorb iron to meet the metabolic demands of ferroptosis.

In our previous work, we discovered that PKCβII acts as a sensor of lipid peroxidation, which is also activated by lipid peroxides during the process of ferroptosis. This activation subsequently phosphorylates ACSL4, promoting the synthesis of PUFAs, thereby creating a positive feedback regulation that enhances ferroptosis[44]. However, the peroxidation of a large amount of PUFAs relies on iron-catalyzed Fenton reactions, suggesting a critical regulatory mechanism underlying the balance between lipid peroxidation and iron metabolism. Although previous studies have revealed the sources of substrates for lipid peroxides during ferroptosis, the understanding of how iron, another

essential requirement for ferroptosis, is adequately supplied remains limited. In this study, we discovered that PKCβII, the sensor of lipid peroxidation linked to ferroptosis, also serves as a critical mediator of active iron uptake, revealing the synergistic mechanisms coordinating the lipid peroxidation and iron supply in the induction of ferroptosis.

AAK1 has been confirmed to facilitate clathrin-mediated endocytosis through the phosphorylation of AP2M1, which enhances its binding affinity to cargo membrane proteins[40–43]. However, it remains unclear whether AAK1 is activated and influences ferroptosis by regulating the endocytosis of TFR1 and iron uptake. Here, we found that PKCβII is a significant regulator of iron metabolism linked to ferroptosis. We demonstrate that PKCβII directly interacts with and phosphorylates AAK1 at S670/T674 dual sites, promoting AAK1-mediated phosphorylation of AP2M1. Thereby, PKCβII enhances the endocytosis of TFR1 and iron uptake, ultimately promoting ferroptosis of cancer cells.

In summary, the PKCβII-AAK1-AP2M1 axis serves as a critical regulatory pathway for extracellular iron uptake during the induction of ferroptosis in tumor cells, providing an exogenous supply of iron that triggers the accumulation of lipid peroxides linked to ferroptosis. We also demonstrate that PKCβII promotes the endocytosis of TFR1 through the phosphorylation of AAK1, thereby facilitating ferroptosis and inhibiting tumor growth in vivo, which presents a potential therapeutic target for ferroptosis in clinical treatment.

## Methods

### Cell lines

Human breast cancer cell MDA-MB-231 and MCF7, human fibrosarcoma cell HT1080, human Non-Small Cell Lung Cancer cell A549 were obtained from American Type Culture Collection. Human squamous carcinoma cell HN6 was obtained from Cellosaurus. Human breast cancer cell CAL51 was obtained from Deutsche Sammlung von Mikroorganismen und Zellkulturen. MDA-MB-231, MCF7, A549, CAL51, and HN6 were cultured in DMEM medium with 7% fetal bovine serum and HT1080 was cultured in RPMI 1640 medium with 7% fetal bovine serum. These cells were cultured at 37 °C in an incubator with a humidified atmosphere of 20% $O_2$ and 5% $CO_2$. All of these cells were cultured in 10 cm plates followed by transferring to 6-well plates before cell-death and lipid-peroxidation measurements. Reagents added to these cells in the article included PKC inhibitors, Go6983 (Selleck, S2911) and enzastaurin (Selleck, S1055); AAK1 inhibitor, SGC-AAK1-1 (Sigma-Aldrich, SML2219); ferroptosis inducer, erastin (TargetMol, T1765); ferroptosis inhibitors Fer-1 (Selleck, S7243), DFO (Selleck, S5742) and NAC (Selleck, S1623); apoptosis inhibitor, Z-VAD-FMK (Selleck, S7023); necroptosis inhibitor, Nec (Selleck, S8037); For cystine deprivation, cells were cultured in DMEM medium without cystine (GIBCO, 21013024) for 12 h.

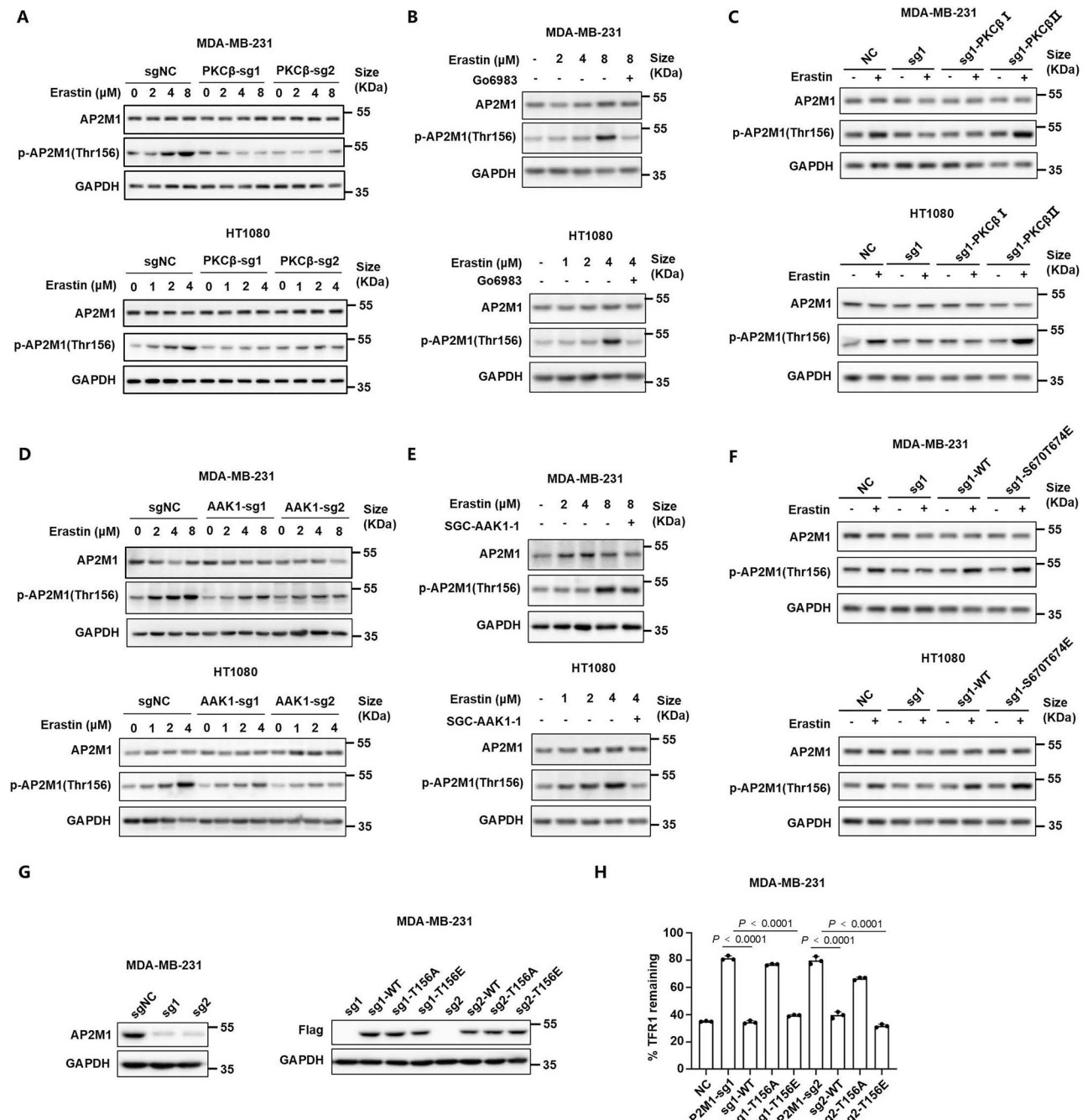

**Fig. 4 | PKCβII-AAK1-AP2M1 pathway is essential for the endocytosis of TFR1 and iron uptake during ferroptosis. A** The phosphorylation levels of AP2M1 in *PKCβ*-knockout MDA-MB-231 (top) and HT1080 (bottom) cells treated with erastin at different concentrations for 12 h. **B** The phosphorylation levels of AP2M1 in MDA-MB-231 (top) and HT1080 (bottom) cells treated with erastin at different concentrations with/without 5 μM Go6983 for 12 h. **C** The phosphorylation levels of AP2M1 in the indicated MDA-MB-231 (top) and HT1080 (bottom) cells treated with/without 10 μM or 5 μM erastin, respectively for 16 h. **D** The phosphorylation levels of AP2M1 in *AAK1*-knockout MDA-MB-231 (top) and HT1080 (bottom) cells treated with erastin at different concentrations for 12 h. **E** The phosphorylation levels of AP2M1 in MDA-MB-231 (top) and HT1080 (bottom) cells treated with erastin at different concentrations with/without 10 μM SGC-AAK1-1 for 12 h. **F** The phosphorylation levels of AP2M1 in the indicated MDA-MB-231 (top) and HT1080 (bottom) cells treated with/without 10 μM or 5 μM erastin, respectively for 16 h. **G** Knockout of *AP2M1* was performed in MDA-MB-231 cells using single guide RNAs (sgRNAs). Plasmids of *AP2M1-WT*, *AP2M1-T156A*, and *AP2M1-T156E* were transfected into *AP2M1*-knockout cells. These cells were verified by immunoblots. **H** Endocytosis assays of TFR1 in the indicated MDA-MB-231 cells treated with 2 μM erastin for 12 h. **A–G** Data are representative of *n* = 3 biologically independent experiments. **H** Data are presented as means ± SD, *n* = 3 biologically independent experiments, unpaired two-tailed Student's *t* test.

## Plasmid constructs

The plasmid containing *PKCβII* (HG10750-CH) gene was purchased from Sino Biological, and plasmids containing *AAK1* (P23548) and *AP2M1* (P23673) genes were purchased from Miaoling Biology. *PKCβI* plasmid was constructed through transferring part of sequence of *PKCβII*. We constructed all plasmids for stably expression in tumor cells with pCDH-Neo as carrier vector. Plasmid expressing fusion protein of AAK1 for in vitro kinase assay was constructed using PGEX-6P-1 vector. Plasmid containing *AAK1* gene for immunoprecipitation from HEK293T was constructed using pcDNA3.1 vector.

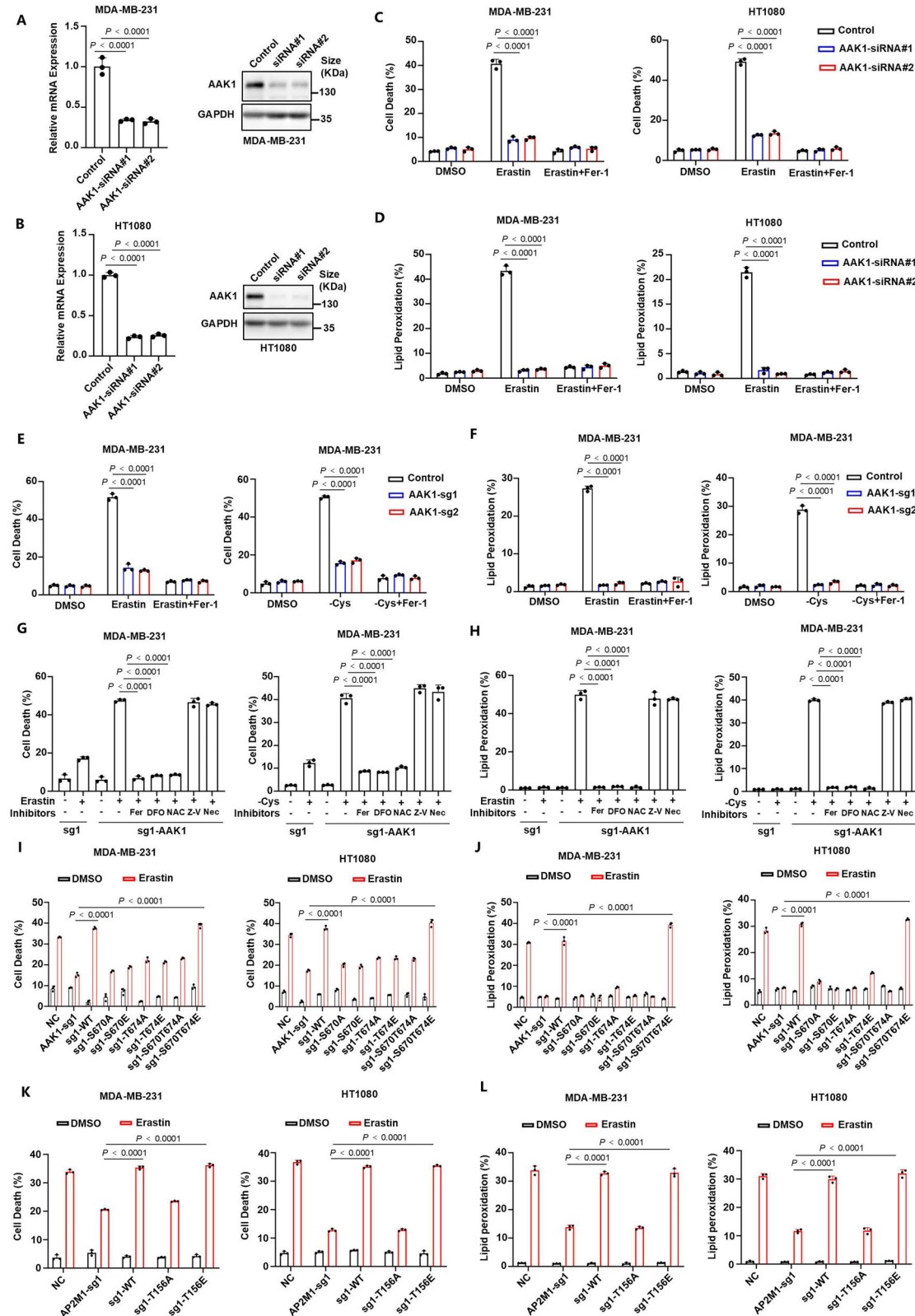

## Gene knockout by Crispr-cas9 system

Knockout of *PKCβ*, *AAK1*, and *AP2M1* were performed by Crispr-cas9 technique. We constructed all the plasmids containing sgRNA for genetic disruption with Lenti-Crispr-V2 vector. Sequences of sgRNA are as follows: sgPKCβ, 5′-TCAGCCCTATGGGAAGTCCG-3′, 5′-GTGACAACCAAGACATTCTG-3′; sgAAK1, 5′-CGGGCGACAGCAGGTCACAG-3′, 5′-TGGCAAAATCATCACTACGA-3′; sgAP2M1, 5′-GATCT TCTGAGCCAGCAGTG-3′, 5′-GAGAGGGTATCAAGTATCGT-3′. These plasmids containing sgRNA sequences were transfected into HEK293T cells with psPAX2 packaging plasmid and pMD2.G VSV-G envelope-expressing plasmid, followed by virus collection in 72 h. MDA-MB-231 and HT1080 cells were infected by the collected virus with 1 µg/ml polybrene and selected with 2 µg/ml puromycin for 3 d.

**Fig. 5 | PKCβII-AAK1-AP2M1 pathway significantly enhances the sensitivity to ferroptosis. A, B** *AAK1* depletion by specific siRNA in MDA-MB-231 (**A**) and HT1080 (**B**) cells and verified by real-time qPCR (left) and western blotting (right). Data are representative of n = 3 biologically independent experiments. **C, D** Cell-death (**C**) and lipid-peroxidation (**D**) measurements for the indicated MDA-MB-231 (left) and HT1080 (right) cells treated with 10 μM or 5 μM erastin, respectively with/without 10 μM Fer-1 for 12 h. **E, F** Cell-death (**E**) and lipid-peroxidation (**F**) measurements for the indicated MDA-MB-231 cells treated with erastin (left) or cystine deprivation (right) with/without Fer-1 for 12 h. erastin, 10 μM; Fer-1, 10 μM. -Cys, cystine deprivation. **G, H** Cell-death (**G**) and lipid-peroxidation (**H**) measurements for the indicated MDA-MB-231 cells treated with erastin (left) or cystine deprivation (right)

with/without various cell death inhibitors for 16 h. erastin, Fer-1 and DFO, 10 μM; -Cys, cystine deprivation; NAC, 5 mM N-acetyl-cysteine; Z-V, 10 μM Z-VAD-FMK; Nec, 2 μM necrostatin-1s. **I, J** Cell-death (**I**) and lipid-peroxidation (**J**) measurements for the indicated MDA-MB-231 (left) and HT1080 (right) cells treated with 10 μM or 5 μM erastin, respectively for 16 h (cell death) or 12 h (lipid-peroxidation). **K, L** Cell-death (**K**) and lipid-peroxidation (**L**) measurements for the indicated MDA-MB-231 (left) and HT1080 (right) cells treated with 10 μM or 5 μM erastin, respectively for 12 h (cell death) or 12 h (lipid-peroxidation). **A–F** Data are presented as means ± SD, n = 3 biologically independent experiments, one-way ANOVA test. **G–L** Data are presented as means ± SD, n = 3 biologically independent experiments, unpaired two-tailed Student's t test.

### Genetic depletion by siRNA

The siRNA mediating genetic depletion of *AAK1* was purchased from GenePharma. Sequences of siRNA are as follows: 5′-CAAGAAUAUU-GUGGGUUACAUUGAU-3′, 5′-GAGCCGUCUCAAGUUUAAACUUACA-3′. The powder of synthetic siRNA was dissolved with DEPC water before transfected into cells cultured in 6-well plate in the addition of 4 μl RNAmate for 2 d. The transfected cells were collected for further experiments.

### RNA extraction and real-time qPCR

Genetic depletion of AAK1 was verified by RT-qPCR. The siRNA transfected cells were planted in a 6-well plate and collected for RNA extraction using RNA Purification Kit (EZBioscience, B0004D). Reverse transcription was performed using miRNA 1st Strand cDNA Synthesis Kit (Vazyme, MR201-01). The sequences of primers established in RT-qPCR are as follows: AAK1, forward primer 5′-AGTGGCTACATCGGAA-GAGTC-3′, reverse primer 5′-AGGCACATTTCATCCCATTGC-3′; GAPDH, forward primer 5′-GGAGCGAGATCCCTCCAAAAT-3′, reverse primer 5′-GGCTGTTGTCATACTTCTCATGG-3′. Real-Time qPCR was performed using ChamQ SYBR qPCR Master Mix (Vazyme, Q311-03).

### Cell death assays

All the cell-death assays were performed by staining cells with propidium iodide (MPbio, 0219545810). Cells were transplanted to a 6-well plate with $3 \times 10^5$ cells per well, followed by treatment with the indicated compounds for 12 h. Then, we collected and washed the cells with 1 × PBS before stained them with 5 μg/ml propidium iodide. Cell death was analysed by flow cytometry with all the dead cells stained by propidium iodide. At least 5000 cells were analysed in each group and all experiments were repeated at least three times.

### Lipid peroxidation assays

All the lipid-peroxidation assays were performed by staining cells with BODIPY 581/591 C11 (Invitrogen, D3861). Cells were transplanted to a 6-well plate with $3 \times 10^5$ cells per well, followed by treatment with the indicated compounds for 12 h. Then, we collected and washed the cells with 1 × PBS before stained them with 5 μM BODIPY 581/591 C11 at 37 °C for 30 min. Lipid peroxidation was analysed by flow cytometry with all cells undergoing lipid peroxidation stained by BODIPY 581/591 C11. At least 5000 cells were analysed in each group and all experiments were repeated at least three times.

### Endocytosis assays

We transplanted cells to a 6-well plate in the presence or absence of erastin for 12 h. Then, cells were collected with TrypLE Express (GIBCO, 12605010), placed on ice and washed by 7%-fetal-bovine-serum-containing RPMI 1640 medium with 30 mM HEPES (also used for antibody dilution). We labelled cells with TFR1-specific antibody (Proteintech, 66180-1-Ig) diluted at a ratio of 1:200 and incubated on ice for 1 h, followed by washing cells to remove unbound antibody. Next, cells were resuspended in RPMI 1640 medium without fetal bovine serum with or with treatment of dynasore (TargetMol,

T1848) and incubated at 37 °C in a water-bat. The incubation time of water-bath was 5 min for A549, HT1080, MDA-MB-231, and MCF7 cells, and 10 min for CAL51 and HN6 cells. These samples should be transferred to ice immediately at the indicated times to stop further endocytosis. Subsequently, cells were washed twice using RPMI 1640 medium containing 7% fetal bovine serum and 30 mM HEPES and labelled with Alexa Flour-488 conjugated anti-mouse secondary antibody (Invitrogen, A11029) diluted at a ratio of 1:400 for 30 min, followed by flow cytometry for TFR1 detection. At least 5000 cells were analysed in each group and all experiments were repeated at least three times in flow cytometry.

### Total and divalent iron assays

We used the Cell Total Iron Colorimetric Assay Kit (Elabscience, E-BC-K880-M-96T) and Cell Ferrous Iron Colorimetric Assay Kit (Elabscience, E-BC-K881-M-96T) for cellular total and divalent iron measurements. Cells were planted in a 10 cm plate with $2 \times 10^6$ cells treated with or without erastin at indicated concentrations for 12 h. Then, cells were collected for further experiments using methods reported in manual. The total and divalent iron level of each group were detected by microplate reader through a standard concentration curve. The reagents for the generation of standard concentration curve need to be prepared for immediate use.

### Liable iron measurement by immunofluorescence

Ferrous iron levels were detected using $Fe^{2+}$-sensitive probe called FerroOrange purchased from DOJINDO (F374-3tubes). FerroOrange is a kind of membrane permeable dye that only strongly binds to ferrous iron but not trivalent iron or other metal ions. It can be detected by a fluorescence microscope at excitation of 543 nm. Firstly, cells were planted to a 1.5 cm dish with $2 \times 10^5$ cells with or without treatment of erastin at indicated concentrations and for indicated time periods. Next, we washed the cells with 1 × PBS to further remove residual compounds. We stained cells with FerroOrange diluted with DMEM medium at a ratio of 1:4000 at 37 °C for 30 min. Subsequently, cells were washed twice with 1 × PBS followed by detection of fluorescence microscope.

### Colocalization experiment by Immunofluorescence

We performed indirect immunofluorescence to establish the colocalization of AAK1 and exogenous PKCβII. Cells were planted to a 1.5 cm dish at a density of $2 \times 10^5$ cells with treatment of erastin at indicated concentrations and for indicated time periods. We washed cells with 1 × PBS twice before fixed them with 4% paraformaldehyde (Biosharp, SJ-BL539A) at room temperature for 20 min and then treated them with 0.25% TritonX-100 (MP Biomedicals, 219485480) for 10 min for membrane permeabilization. Cells were blocked with 4% BSA at room temperature for 1 h and then incubated with primary antibodies at a ratio of 1:200: anti-AAK1 (CST, 79832), anti-Flag (CST, 14793S). Next, cells were labelled with Alexa Flour-488 (green light) and Alexa Flour-594 (red light) conjugated secondary antibody at a ratio of 1:400 after being washed twice with 1 × PBST. Nuclear was stained by Hoechst

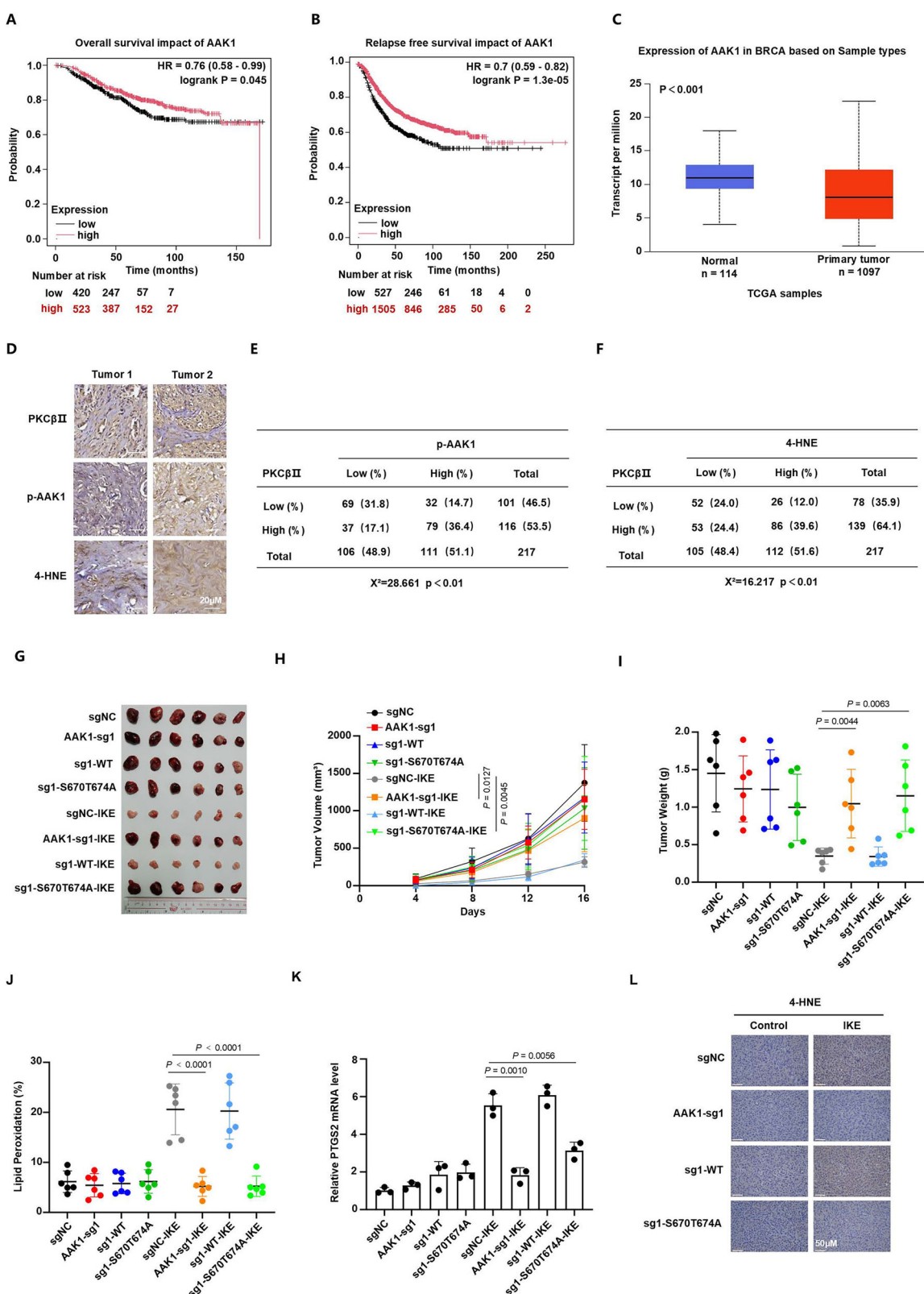

Nature Communications | (2026)17:819                                                                                                    12

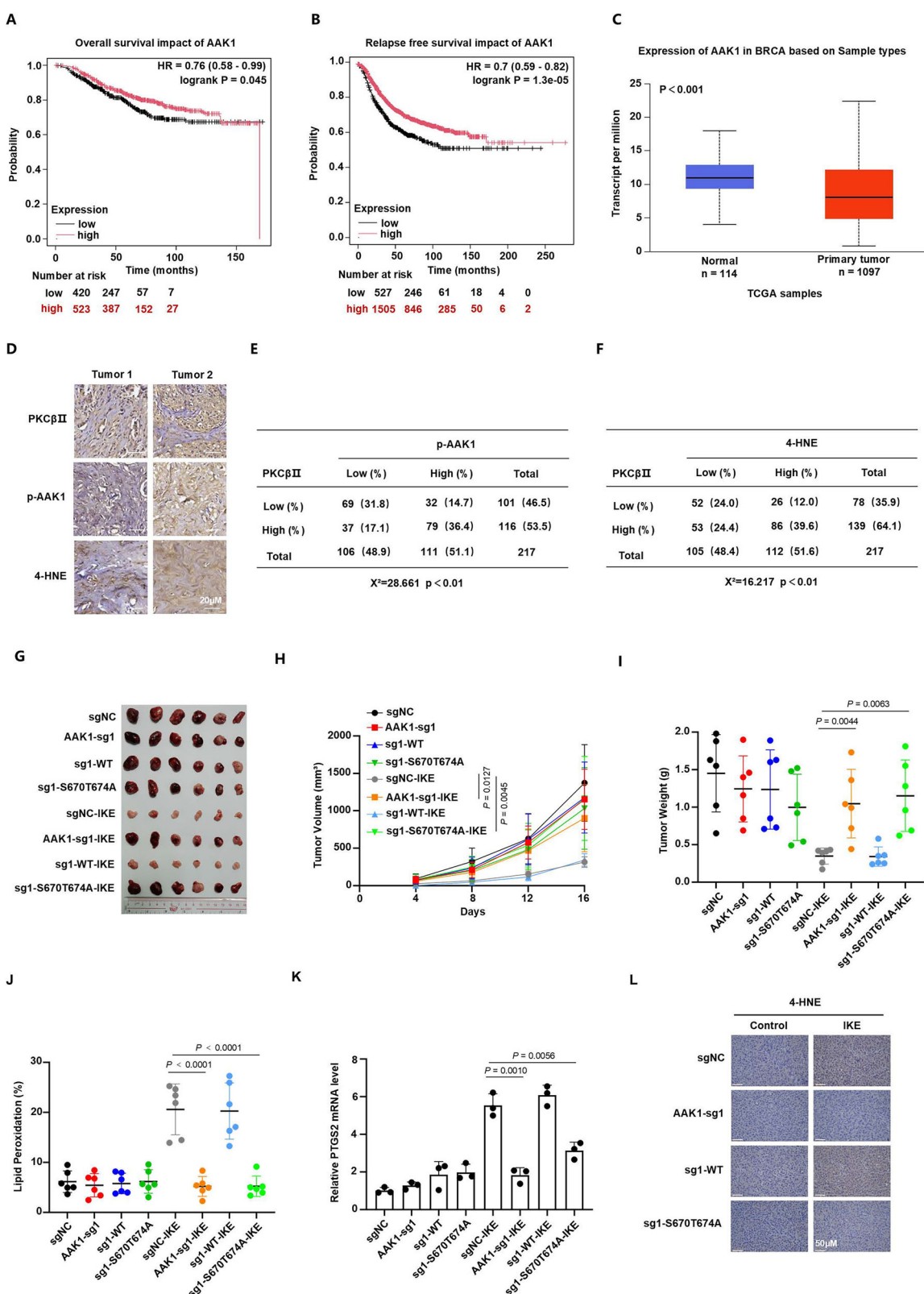

33342 (Beyotime, C1022) for 5 min. Images were acquired with a fluorescence microscope (ZEISS LSM880 with fast Airyscan).

**In vitro kinase assay**

AAK1 fusion proteins were extracted through GST purified system. The plasmid containing sequence of *AAK1* gene was constructed with PGEX-6P-1 as carrier vector and expressed in BL21 *Escherichia coli*,

followed by protein extraction using GST beads (Beyotime, P2138). In addition, AAK1 protein was also immunoprecipitate-isolated using an anti-Flag antibody from HEK293T cells that expressed plasmids of Flag-AAK1-WT or Flag-AAK1-S670/T674Ala whose carrier vector was pcDNA3.1. AAK1 fusion proteins or immunoprecipitated-isolated proteins were incubated with human recombinant activated PKCβII kinase (Abcam, ab60841) in 1 × kinase buffer (CST, 9802S) in the addition of

**Fig. 6 | Activation of PKCβII-AAK1-AP2M1 pathway inhibits tumor growth through the induction of ferroptosis in vivo. A**, **B** Kaplan–Meier analysis of overall survival (**A**) and relapse free survival (**B**) related to the expression of AAK1 for breast cancer patients. **C** Relative expression analysis of AAK1 in primary tumor tissues of breast cancer or normal tissues based on TCGA samples. Normal group: minima 4.048; maxima 17.99; median 10.928; Q1 9.372; Q3 12.878. Primary tumor group: minima 0.784; maxima 22.364; median 8.099; Q1 4.903; Q2 12.135. **D** Representative images of immunohistochemical analysis of tissue chips containing 217 human TNBC specimens using anti-PKCβII, anti-p-AAK1 (S670/T674) and anti-4-HNE antibodies. **E**, **F** Correlation analysis between PKCβII and the

phosphorylation level of AAK1 (**E**) or 4-HNE (**F**) based on tissue chips containing 217 human TNBC specimens. Statistical analysis was performed by Pearson Chi-square ($\chi^2$) test. **G–I** Tumor volume and weight of xenograft tumors formed by the indicated MDA-MB-231 cells treated with DMSO or IKE. **J** Lipid-peroxidation measurement for tumor cells isolated from the indicated tumors. **K** Relative mRNA level of *PTGS2* detected by RT-qPCR. These samples were tumor cells isolated from the indicated tumors. **L** Representative immunohistochemical images of 4-HNE in the indicated tumor tissues. **H** Data are presented as means ± SD, $n = 6$ biologically independent samples, two-way ANOVA test. **I–K** Data are presented as means ± SD, $n = 6$ independent samples, unpaired two-tailed Student's $t$ test.

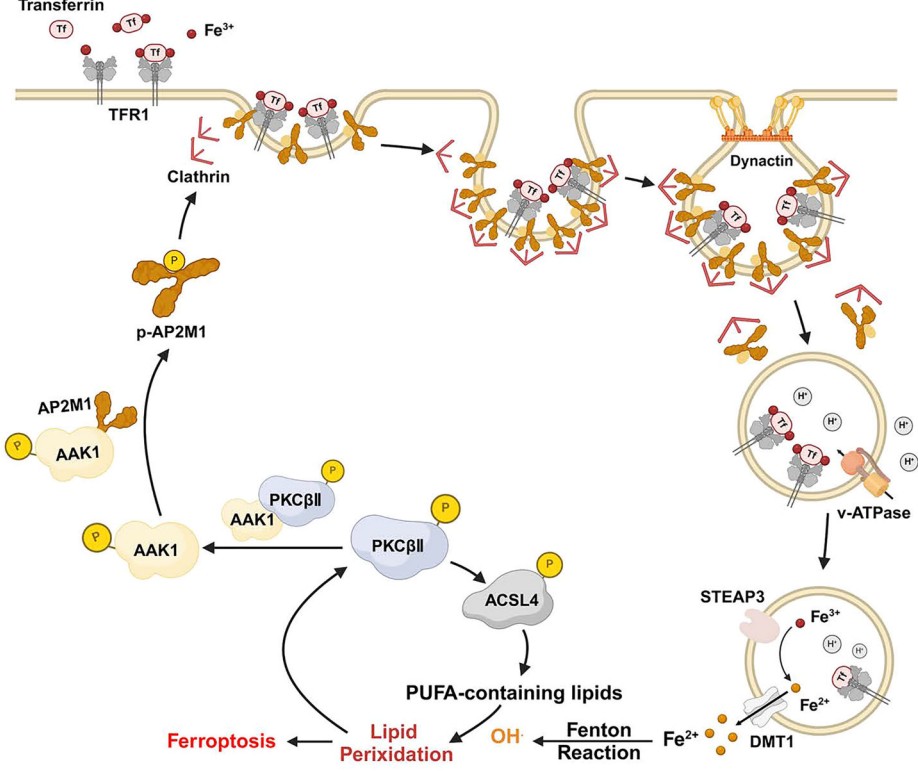

**Fig. 7 | Schematic of the PKCβII-AAK1-AP2M1 pathway.** Schematic of the PKCβII-AAK1-AP2M1 pathway. PKCβII facilitates the phosphorylation of AAK1, which subsequently causes elevated phosphorylation of AP2M1. These phosphorylation cascade reactions enhance clathrin-mediated endocytosis of TFR1 and increase

cellular iron levels, thus providing adequate ferrous iron for lipid peroxidation and promoting ferroptosis of tumor cells. This figure was created in BioRender. Lichao, L. (2025) https://BioRender.com/vryk8z5.

400 μM ATP (CST, 9804S) with or without 0.5 μl λ-phosphatase at 30 °C for 1 h. Negative control was performed by the reaction of AAK1 proteins or PKCβII kinase alone under the same conditions. The reaction was stopped with 20 μl 3 × SDS-PAGE buffer. In vitro kinase assay was finally verified by western blotting.

### Immunoblotting and immunoprecipitation

To perform immunoblotting, proteins were extracted from cells lysed with RIPA containing phosphatase or protease inhibitors, followed by concentration detection. 10 μg of protein was separated using SDS–PAGE at 80 volts for 2 h and transferred to a PVDF membrane at 130 mA for 3 h. To perform immunoprecipitation, cells were lysed with Pierce immunoprecipitation lysis buffer containing phosphatase or protease inhibitors, followed by concentration detection. Lysate containing 1 mg total protein was incubated with 2 μg primary antibody at 4 °C overnight. On the second day, target protein was isolated from protein-antibody mixture through the addition of 30 μl of Pierce protein A/G agarose beads (MCE, HY-K0202) and incubated at 4 °C for 2 h. Next, the beads were washed four times with cold Pierce

immunoprecipitation lysis buffer and heated to 100 °C for 10 min under tough shake in the addition of 60 μl 2 × SDS buffer. The results of immunoprecipitation were detected by western blotting. Antibodies mentioned above were as follows: GAPHD (Proteintech, 60004-1-Ig, 1:5000), TFR1 (Proteintech, 66180-1-Ig, 1:2000), FTH1 (HUABIO, ET1705-55, 1:1000), FTL (HUABIO, EM1707-17, 1:1000), DMT1 (HUABIO, ER1907-55, 1:1000), FPN1 (HUABIO, HA601178, 1:1000), PKCβ (Proteintech, 12919-1-AP, 1:2000), PKCβI (Santa Cruz, sc-8049, 1:500), PKCβII (Santa Cruz, sc-13149, 1:500), AAK1 (CST, 79832T, 1:2000), AP2M1 (Abclonal, A11070, 1:2000), Flag (Abcepta, AP1013A, 1:1000), p-Ser/Thr (Abcam, ab17464, 1:3000), p-AP2M1(Thr156) (Abcam, ab109397, 1:3000).

### Xenograft tumor model

This study complied with all relevant ethical regulations. All procedures involving mice and experimental protocols were approved by the Institutional Animal Care and Use Committee of Sun Yat-sen University Cancer Center. 5-week-old female BALB/c nude mice were obtained from Sun Yat-sen University. All mice were kept under

specific-pathogen-free conditions in the Animal Facility of Sun Yat-sen University Cancer Center. The indicated MDA-MB-231 cells were suspended and counted followed by injecting into mice with the number of $2 \times 10^6$ cells. Administration was performed in the fourth day after the injection when the tumors reached 50–100 mm³. IKE was dissolved in solvent containing 5% DMSO, 40% PEG300, 5% Tween 80 and 50% saline and then intraperitoneally injected to mice at a dose of 40 mg/kg. In addition, BSO was dissolved in saline and intraperitoneally injected to mice at a dose of 0.6 g/kg. IKE and BSO were administrated once every 4 days. Lipro-1 was dissolved in PBS, followed by intraperitoneally daily injection at a dose of 10 mg/kg. Nude mice were treated with IR of 10 Gy in the fourth day after the inoculation. The tumor volumes were measured as indicated in the corresponding figures with the formula volume = length × width² × 0.5. The weight of tumor tissues was measured after mice were killed. Part of tumor tissues was isolated and used for lipid-peroxidation measurement and Immunohistochemistry. Tumor size less than 2000 mm³ was permitted by ethics committee and all tumors met the requirement.

## Statistical analysis

Statistical analyses were conducted using the GraphPad Prism (version 8.0) software. The results are presented as the mean ± SD of three or six biologically independent experiments or samples. The results were analysed by an unpaired Student's $t$-test, one-way ANOVA or two-way ANOVA with the Dunnett's multiple comparisons test, or Pearson Chi-square ($\chi^2$) test. All statistical tests were two-sided and $P < 0.05$ was considered statistically significant.

## Immunohistochemistry

Human TNBC tissue chips and tumor tissues isolating from mice were fixed with 4% paraformaldehyde overnight. These samples were then embedded with paraffin and sectioned into slices. Next, we immersed the tissue slices in EDTA citrate buffer and completed antigen retrieval with microwave, followed by blocking them with 4% BSA for 30 min. Incubation of primary antibodies against PKCβII, p-AAK1, and 4-HNE were performed at 4 °C overnight. After that, the sliced samples were incubated with HRP-conjugated secondary antibody for 30 min. In addition, we used Haematoxylin for counterstaining. Images were acquired using a digital pathology slide scanner (KFBIO, KF-PRO-020) and analysed by HALO software (Indica Labs, version 3.6.4134).

## Phosphoproteomic analysis

Phosphoproteome was performed according to methods below, which has been confirmed by other study[53,54]. To extract total proteins, we cultured wild-type and *PKCβ*-KO MDA-MB-231 cells in 15 cm plates with an amount of $6 \times 10^6$. Each group was repeated with 3 independent biological samples. These cells were lysed with lysis buffer supplemented with phosphatase inhibitors and protease inhibitors for 15 min on ice, followed by high-intensity sonication at 4 °C. We then collected the lysate by centrifugation at $12,000 \times g$ for 15 min at 4 °C, followed by concentration detection using BCA kit. High pH reversed phase HPLC were used for fractionation. Samples were firstly fractionated into 80 fractions, followed by combining into 12 fractions and drying by vacuum centrifugation. Next, the phosphorylated peptides were enriched using TiO2 for further mass spectrometry. Raw phosphoproteome MS/MS data were analysed using the MaxQuant search engine (v.1.5.3.30).

## Reporting summary

Further information on research design is available in the Nature Portfolio Reporting Summary linked to this article.

## Data availability

Overall survival and relapse free survival related to the expression of *AAK1* for breast cancer patients were performed by Kaplan–Meier analysis (https://kmplot.com/analysis/). The relative expression of *AAK1* in primary tumor tissues of breast cancer or normal tissues was analysed by using The Cancer Genome Atlas (TCGA, https://www.cancer.gov/ccg/research/genome-sequencing/tcga). GOBP gene set containing anti-ferroptosis genes (https://www.gsea-msigdb.org/gsea/msigdb/human/geneset/GOBP_FERROPTOSIS) was used for the correlation analysis of the expression of *AAK1* based on TCGA dataset and ICGC-BC dataset. The raw data of phosphoproteome that detected the phosphorylation levels of proteins in *PKCβ*-knockout MDA-MB-231 cell lines during ferroptosis is supplied in Supplementary Data 1. Source data are provided with this paper.

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

## Acknowledgements

This study was supported by the Science Fund for Creative Research Groups of the National Science Foundation of China [grant nos. 82321003 (to X.-F.Z.)], the Natural Science Foundation of China [grant nos. 82130079 (to X.-F.Z.), 82341029 (to X.-F.Z.), 82272708 (to H.-L.Z.), 82203205 (to Y.X.), 82203128 (to X.-Y.Z.), 82203753 (to Q.-Q.L.)].

## Author contributions

X.-F.Z., H.-L.Z., and R.D. conceived the idea. L.-C.L., Z.-P.Y., Y.X., X.-Y.Z., Q.-Q.L., Y.-Q.G., H.-L.W., Z.-L.L., and Y.-H.C. performed most experiments. X.-Y.Z., L.-Y.W., G.-K.F., D.Y., S.L., B.-X.H., J.-H.T., Y.-F.Z., and J.L. performed the animal experiments. Y.-H.C. performed the immunohistochemical staining. H.-L.Z., R.D., and X.-F.Z. wrote the manuscript. All co-authors have seen and approved the manuscript.

## Competing interests

The authors declare no competing interests.
