## [Transparent Peer Review file · Nature Communications]

AAK1 activation-mediated iron trafficking drives ferroptotic cell death

Corresponding Author: Professor Xiao-Feng Zhu

Version 0:

Reviewer comments:

Reviewer #1

(Remarks to the Author)

Li et al. identify PKC β II phosphorylation and activation of AP2-associated 38 protein kinase 1 (AAK1) as a regulatory step in transferrin receptor (TFR1)-dependent endocytosis during ferroptosis. Lipid peroxides and iron accumulation are prerequisites for the induction of ferroptosis. In cells, iron is stored in a complex with ferritin, and the degradation of ferritin results in increased iron levels, potentially leading to ferroptosis. Paradoxically, the group and others observed that induction of ferroptosis is accompanied by an increase in ferritin. Thus, the uptake of iron might be needed to initiate ferroptosis. To explain this, the authors investigate the regulation of the transferrin-mediated iron uptake through transferrin receptor 1 (TFR1) and endocytosis during erastin-induced ferroptosis. Based on their previous work, the authors focus on PKC β in this process. Using phosphoproteomics, they identify AAK1 as a differentially regulated substrate of PKC β and implicate it in TFR1 uptake. The authors demonstrate that AAK1 and PKC β II interact and that AAK1 is a substrate of PKC β II. Furthermore, they investigate the phosphorylation of a known AAK1 substrate, AP2M1, that functions in clathrin-mediated endocytosis and can link AP2M1 to PKC β II, TFR1 endocytosis and the regulation of ferroptosis. Furthermore, they demonstrate that AAK1 depletion inhibits ferroptosis and lipid peroxidation in a phosphorylation-dependent manner. Finally, they evaluate the significance of their findings in a xenograft mouse model of breast cancer.

This is a well-designed study using a core set of assays to elucidate a signaling pathway involved in iron uptake and regulating ferroptosis. Appropriate controls are included for most assays. The section demonstrating the interaction between PKC β II and AAK1 and the phosphorylation of AAK1 by PKC β II following the discovery of AAK1 as a substrate of PKC β by phosphoproteomics is very detailed, lasting from Fig. 2A to their validation in Fig. 3F/G.

Comments:

1. The authors show in Figure 1 increased endocytosis of TFR1 upon erastin-induced ferroptosis. As a control, the authors should block endocytosis. Is there a synergistic effect of blocking endocytosis and depleting PKC β ? In Figure 2, the authors use dynasore in combination with AAK1 KO. Why not here?
2. The finding that only one of the PKC β transcripts rescues the effect of PKC β depletion is interesting and consistent with the authors' previous work. However, are both PKC β transcripts expressed and translated? What is the relative expression of both proteins? Can increased expression of PKC β I eventually rescue the depletion effect? What antibodies are used for the detection of PKC β I and II proteins by WB? Does PKC inhibition have a similar effect compared to PKC β KO?
3. For the phosphoproteomics data, I could not find a corresponding table. Also, the raw data does not seem to be deposited in ProteomeXchange/Massive. It is stated that phosphorylation levels were observed, but only proteins are indicated. Which sites were detected on ACSL4 and AAK1? Is the phosphoproteomics data corrected for changes in protein abundance? Are the cells used in this analysis PKC-KO cells (as indicated in line 190) or PKC β KO cells? What other proteins are regulated?
4. Does AAK1 specifically interact with PKC β II? Or does it also interact with PKC β I? To substantiate the claim that PKC inhibitors substantially decrease the interaction, quantification, and statistical testing of the Western blot (WB) results should be included. Conversely, the increase in colocalization of AAK1 and PKC β II upon erastin treatment should be quantified. Did all kinase reactions shown in Figure 2 J and K include ATP? In line 292, it is stated that knock-in cell lines for the phosphorylation site mutants were generated. It is necessary to clarify that these are exogenous expression plasmids, not endogenous knock-ins. Extended data 5G indicates that pSer/Thr is the used antibody. Is that the same antibody they used in Extended Data 5A-E or the validation of the AAK1 S670/T407 antibody they generated? Does the antibody only work on purified or IPed protein? Does phosphorylation of AAK1 enhance its activity in vitro? Are the sites in any specific protein domains?

5. Why did the authors use siRNA and gRNAs to deplete cells of AAK1 in Figure 5?
6. In Figure 6C, is the difference in expression of AAK1 in primary tumor tissues versus normal tissues significant? The negative correlation in Extended Data 9A/B is weak, which should be mentioned in the text. Is there a difference in survival in patients with high and low expression of PKC β II or AP2M1 in the same dataset as used for Fig. 6A/B?
7. At what point after inoculation with cells were the mice treated with IKE?

Minor points:

1. The following statement starting in line 110 needs a citation: "We observed that the levels of ferritin were generally increased during ferroptosis as previously reported". The same applies to the reference to prior literature in line 123.
2. The model is nice, but additional details on the biochemical reactions would enhance it.

Reviewer #2

(Remarks to the Author)

The authors investigate how cells regulate iron uptake during ferroptosis, an open point in the field. They identify a kinase that mediates transferrin receptor endocytosis through the activation of AAK1. They show that AAK1 phosphorylates AP2M1 which increases the endocytosis of iron and promotes ferroptosis in a technically sound and mechanistic way. However, there are significant concerns about novelty, as the AAK1-AP2M1 regulation of endocytosis is well described. In addition, translation and human modeling are absent, reducing our enthusiasm for this paper.

Major:

1. In Figures 1A, 1D and 1E, the authors confirmed that erastin treatment induced FTH1, and intracellular iron. It could be due to either the increased uptake or impaired export of iron. Therefore, the expression of major iron transporters including TFR1, DMT1, FPN1 should be tested. Meanwhile, the recycling of TFR1 should be tested.
2. In Figure 1C, almost no endocytosis was detected in CAL51 and HN6 cells, indicating that there were some technical issues. Probably it was due to only 5-min incubation at 37 degrees. Extending the incubation time may help.
3. In Figure 2A, the authors did phosphoproteomic analysis. In this analysis, many targets were found, including AAK1. AAK1 was not the most dramatically changed gene. Therefore, the authors need to provide a better rationale for why AAK1 stands out.
4. Figure 4A, the variation of GAPDH is too dramatic, which dampens the reliability of the data. For example, the increase of p-AP2M1 with erastin treatment could be caused by uneven protein loading.
5. Figures 4C and 4F, erastin treatment did not induce p-AP2M1 in NC cells, which is controversial to previous data.
6. Figure 4F, WT AAK1 did not restore the phosphorylation of AP2M1.
7. Figures 5I and 5K, only 20-30% cells died after erastin treatment, which is too low for a reliable conclusion.
8. Figure 6C, no significance was shown in the figure. If the data are not significant, the conclusion in rows 446-448 is not reasonable.
9. Figures 6E and 6F, the correlation analysis is too weak.

Minor:

1. Figure 1D and 1E should be moved before endocytosis assay (1B and 1C), because 1D and 1E together with 1A demonstrated the increase of intracellular iron.
2. Extended Figure 1F, please indicate whether the cells were treated with erastin. If so, please specify the concentration and incubation time of the treatment.
3. Row 230-233, PIK inhibitors decreased PKC β II-AAK1 interaction under erastin treatment. This does not suggest that PKC β II-AAK1 complex is involved in ferroptosis induction. The author over-interpreted the data.
10. Figures 2J and 2K are suggested to move to Figure 3. Figures 2J, 2K and 3 together confirms the phosphorylation of AAK1 by PKC β II.
11. Row 282, the function of Go6983, enzastaurin needs to be explained.
12. Row 307-308, S670/T674 needs to be emphasized. Meanwhile, in Figures 3J and 3K, it is suggested to emphasize p-S670/T674 in the figures, even though this has been mentioned in the figure legends.
13. Row 420, data only suggest that PKC β II-AAK1-AP2M1 pathway affects ferroptosis sensitivity but not the induction of ferroptosis.

Version 1:

Reviewer comments:

Reviewer #1

(Remarks to the Author)

The authors have mostly answered my questions. Their comment that the MS raw data will be uploaded soon after revision is not satisfactory; it needs to be completed and checked before final evaluation.

Reviewer #2

(Remarks to the Author)

The authors have addressed previous comments.

Referee#1

(Remarks to the Author)

Li et al. identify PKC β II phosphorylation and activation of AP2-associated protein kinase 1 (AAK1) as a regulatory step in transferrin receptor (TFR1)-dependent endocytosis during ferroptosis. Lipid peroxides and iron accumulation are prerequisites for the induction of ferroptosis. In cells, iron is stored in a complex with ferritin, and the degradation of ferritin results in increased iron levels, potentially leading to ferroptosis. Paradoxically, the group and others observed that induction of ferroptosis is accompanied by an increase in ferritin. Thus, the uptake of iron might be needed to initiate ferroptosis. To explain this, the authors investigate the regulation of the transferrin-mediated iron uptake through transferrin receptor 1 (TFR1) and endocytosis during erastin-induced ferroptosis. Based on their previous work, the authors focus on PKC β in this process. Using phosphoproteomics, they identify AAK1 as a differentially regulated substrate of PKC β and implicate it in TFR1 uptake. The authors demonstrate that AAK1 and PKC β II interact and that AAK1 is a substrate of PKC β II. Furthermore, they investigate the phosphorylation of a known AAK1 substrate, AP2M1, that functions in clathrin-mediated endocytosis and can link AP2M1 to PKC β II, TFR1 endocytosis and the regulation of ferroptosis. Furthermore, they demonstrate that AAK1 depletion inhibits ferroptosis and lipid peroxidation in a phosphorylation-dependent manner. Finally, they evaluate the significance of their findings in a xenograft mouse model of breast cancer. This is a well-designed study using a core set of assays to elucidate a signaling pathway involved in iron uptake and regulating ferroptosis. Appropriate controls are included for most assays. The section demonstrating the interaction between PKC β II and AAK1 and the phosphorylation of AAK1 by PKC β II following the discovery of AAK1 as a substrate of PKC by phosphoproteomics is very detailed, lasting from Fig. 2A to their validation in Fig. 3F/G.

Comments:

1. The authors show in Figure 1 increased endocytosis of TFR1 upon erastin-induced

ferroptosis. As a control, the authors should block endocytosis. Is there a synergistic effect of blocking endocytosis and depleting PKC β ? In Figure 2, the authors use dynasore in combination with AAK1 KO. Why not here?

Response: Thank you for the helpful suggestion. In this study, we blocked endocytosis of TFR1 with dynasore, an endocytosis inhibitor which competitively blocks the GTPase activity of dynamin (**Fig. 2E, F; Extended Data Fig. 1K**). The results showed that the recovery effect of TFR1 endocytosis by exogenous expression of PKC β II or AAK1 was blocked by dynasore, confirming the detection method of TFR1 endocytosis was effective. To strictly confirm that the endocytosis of TFR1 was enhanced during erastin-induced ferroptosis, we provided new data in the revised manuscript that endocytosis of TFR1 was inhibited by dynasore in a panel of cancer cell lines treated with erastin (**Figure. 1E**).

About whether there is a synergistic effect of blocking endocytosis and depleting PKC β , we performed endocytosis assay in PKC β - or AAK1-knockout cells treated with dynasore during erastin-induced ferroptosis. The results revealed that there was no extra enhancement of endocytosis inhibition with the combination of PKC β or AAK1 knockout and dynasore, displaying no synergistic effect of blocking endocytosis and depleting PKC β or AAK1 (**Extended Data Fig. 1F; Extended Data Fig. 2C**).

Figure. 1E

Figure. 1E. (E), Endocytosis assays of TFR1 were performed in various cancer cell lines treated with erastin at different concentrations for 12h. A549, 4 μ M; CAL51, 20 μ M; HN6, 20 μ M; HT1080, 1 μ M; MDA-MB-231, 2 μ M; MCF7, 20 μ M. Statistical analysis was performed using an unpaired two-tailed Student's t-test, * $p < 0.01$.

Figure. 2E, F

Figure. 1E, F. (E), AAK1-knockout MDA-MB-231 cells were transfected with sgRNA-resistant AAK1 and verified by western blotting. Endocytosis assays of TFR1 were performed in the indicated cells treated with erastin with/without dynasore. erastin, 2 μ M for 12h; dynasore, 150 μ M for 1h. **(F)**, AAK1-knockout HT1080 cells were transfected with sgRNA-resistant AAK1 and verified by western blotting. Endocytosis detection of TFR1 were performed in the indicated cells treated with erastin with/without dynasore. erastin, 1 μ M for 12h; dynasore, 150 μ M for 1h. Statistical analysis was performed using an unpaired two-tailed Student's t-test, * p <0.01.

Extended Data Fig. 1F, K

Extended Data Fig. 1F, K. (F), Endocytosis assays of TFR1 in the indicated MDA-MB-231 cells treated with 4 μ M erastin for 12h. Dynasore, 150 μ M for 1h. **(K)**, Endocytosis assays of TFR1 in the indicated MDA-MB-231 (left) and HT1080 (right) cells treated with 2 μ M or 1 μ M erastin for 12h respectively. dynasore, 150 μ M for 1h. Statistical analysis was performed using an unpaired two-tailed Student's t-test, * p <0.01.

Extended Data Fig. 2C

Extended Data Fig. 2C. (C), Endocytosis assays of TFR1 in the indicated MDA-MB-231 cells treated with 4 μ M erastin for 12h. Dynasore, 150 μ M for 1h. Statistical analysis was performed using an unpaired two-tailed Student's t-test, * $p < 0.01$.

2. The finding that only one of the PKC β transcripts rescues the effect of PKC β depletion is interesting and consistent with the authors' previous work. However, are both PKC β transcripts expressed and translated? What is the relative expression of both proteins? Can increased expression of PKC β I eventually rescue the depletion effect? What antibodies are used for the detection of PKC β I and II proteins by WB? Does PKC inhibition have a similar effect compared to PKC β KO?

Response: Thank you for raising a good point. As we described in the manuscript, PKC β has two transcripts, PKC β I and PKC β II. The latter includes a 50-amino-acid hydrophobic sequence at its C-terminus, which may contribute to its enhanced activation and improved membrane localization during ferroptosis. Both of the transcripts were expressed and translated in a similar level. In the study, we knocked out PKC β using single guide RNA (sgRNA) against both of the transcripts, followed by transfecting sgRNA-resistant plasmids of PKC β I and PKC β II, which could be detected by different antibodies (**Figure. 1J, K**). The antibody using for immunoblots of PKC β I was purchased from Santa Cruz (sc-8049), while PKC β II was purchased from Santa Cruz (sc-13149). In our previous research, we found that only PKC β II rescued the sensitivity to ferroptosis in PKC β -knockout cells. In this manuscript, we also revealed that PKC β II, but not PKC β I, almost completely reversed the inhibitory effect of PKC β knockout on the endocytosis of TFR1 during erastin-induced ferroptosis (**Figure. 1J, K**). In addition, Go6983 or enzastaurin, inhibitors targeting

all subtypes of PKC, were found to inhibit the phosphorylation of AAK1 and interaction between PKC β II and AAK1, revealing a similar effect compared to PKC β KO (Figure. 3A, B; Extended Data Fig. 4C, D).

Figure. 1J, K

Figure. 1J, K. Plasmids of PKC β I or PKC β II were transfected into PKC β -knockout MDA-MB-231 (**J**) and HT1080 (**K**) cells respectively and verified by western blotting. Endocytosis assays of TFR1 were performed in these cells treated with 2 μ M or 1 μ M erastin for 12h respectively. Statistical analysis was performed using an unpaired two-tailed Student's t-test, *p < 0.01.

Fig. 3A, B

Fig. 3A, B. (A), Total phosphorylation levels of AAK1 in MDA-MB-231 cells treated with erastin at different concentrations (left) and for different time periods (right) as indicated, with/without Go6983 by Immunoblots. Go6983, 5 μ M. (B), Total phosphorylation levels of AAK1 in MDA-MB-231 cells treated with erastin at different concentrations (left) and for different time periods (right) as indicated, with/without enzastaurin by Immunoblots. Enza, 5 μ M enzastaurin.

Extended Data Fig. 4C, D

Extended Data Fig. 4C, D. Endogenous AAK1 was immunoprecipitated from MDA-MB-231 (left) and HT1080 (right) cells treated with erastin at different concentrations for 12h with/without Go6983 (C) or enzastaurin (D) followed by immunoblots using a PKCβII-specific antibody. G06983, 5μM; Enza, 5μM enzastaurin.

3. For the phosphoproteomics data, I could not find a corresponding table. Also, the raw data does not seem to be deposited in ProteomeXchange/Massive. It is stated that phosphorylation levels were observed, but only proteins are indicated. Which sites were detected on ACSL4 and AAK1? Is the phosphoproteomics data corrected for changes in protein abundance? Are the cells used in this analysis PKC-KO cells (as indicated in line 190) or PKC KO cells? What other proteins are regulated?

Response: The raw data of phosphoproteomics will be uploaded soon after the revision. We performed the phosphoproteomic analysis with PKCβ-knockout cells during erastin-induced ferroptosis (Figure. 2A). The description “PKC-knockout cells” in line 190 was corrected to “PKCβ-knockout cells”, consistent to the figure legend (Please refer to highlighted sentence at Page 8 in the revised manuscript). The phosphorylation score was corrected for changes in protein abundance. In our previous research, we found that ACSL4 was phosphorylated by PKCβII at Thr328 during ferroptosis, which was also detected in the phosphoproteomics. Besides,

AAK1 was potentially phosphorylated at S670 and T674. Moreover, FOXK1, which was reported to promote the intercellular spread of ferroptosis, was also found in the phosphoproteomics.

Fig. 2A

Fig. 2A. Phosphorylation level and gene score for individual genes were analyzed in the phosphoproteomic experiment using three individual wild type and PKC β -KO MDA-MB-231 cells. Potential genes exhibiting lower phosphorylation level have a negative score and genes exhibiting higher phosphorylation level have a positive score. Red highlights AAK1. Blue highlights ACSL4.

4. Does AAK1 specifically interact with PKC β II? Or does it also interact with PKC β I?

To substantiate the claim that PKC inhibitors substantially decrease the interaction, quantification, and statistical testing of the Western blot (WB) results should be included. Conversely, the increase in colocalization of AAK1 and PKC β II upon erastin treatment should be quantified. Did all kinase reactions shown in Figure 2J and K include ATP? In line 292, it is stated that knock-in cell lines for the phosphorylation site mutants were generated. It is necessary to clarify that these are exogenous expression plasmids, not endogenous knock-ins. Extended data 5G indicates that pSer/Thr is the used antibody. Is that the same antibody they used in Extended Data 5A-E or the validation of the AAK1 S670/T407 antibody they generated? Does the antibody only work on purified or IPed protein? Does phosphorylation of AAK1 enhance its activity? Are the sites in any specific protein domains?

Response: Thank you for the helpful suggestion. We supplemented data that explored the interaction of PKC β I and AAK1. Co-IP experiments revealed a weak interaction between PKC β I and AAK1, which was not enhanced during erastin-induced

ferroptosis, suggesting that PKC β I may not be involved in the ferroptotic process (**Extended Data Fig. 4E, F**). Based on recommendations, Co-IP experiments and immunofluorescent staining were quantified by statistical testing (**Fig. 2H, I; Extended Data Fig. 4G-J**). ATP was included in All kinase reactions shown in Figure 2J and K, which was explained in the figure legends (Fig. 2J, K has been adjusted to **Fig. 3E, F**). In addition, the description “knock-in cell model” in line 292 was corrected to “AAK1 exogenous expression cell model” (Please refer to highlighted sentence at Page 13 in the revised manuscript). About the phospho-antibodies we used, description is as follows. There were two phospho-antibodies used in Extended data 5G, “pSer/Thr” was the same antibody used in Extended Data 5A-E, which functions as a universal antibody of phosphorylated Ser/Thr, while “p-AAK1” was the AAK1 S670/T674 antibody we generated, which only works on purified or IPed AAK1 protein. As the manuscript revealed, AAK1 was phosphorylated and activated by PKC β II at S670/T674 dual sites in vitro (**Fig. 3K**). It was showed that the phosphorylation of AP2M1, a downstream protein of AAK1, was inhibited during ferroptosis in AAK1-knockout cells or treatment with AAK1 inhibitor (**Fig. 4D, E**). These results indicated that phosphorylation levels of AAK1 contributed to its activity and function during ferroptosis. However, the function of protein domain containing these sites were poorly understood.

Extended Data Fig. 4E, F

E

F

Extended Data Fig. 4E, F. Endogenous AAK1 was immunoprecipitated from MDA-MB-231 (**E**) and HT1080 (**F**) cells treated with erastin at different concentrations for 12h, followed by immunoblots using a PKC β I-specific antibody to establish the interaction of endogenous AAK1 with endogenous PKC β I.

Fig. 2H, I

H

Fig. 2H, I. (H), Endogenous PKCβII was immunoprecipitated from MDA-MB-231 (left) and HT1080 (right) cells treated with erastin at different concentrations for 12h, followed by immunoblots using a AAK1-specific antibody to establish the interaction of endogenous PKCβII with endogenous AAK1. **(I)**, Endogenous AAK1 was immunoprecipitated from MDA-MB-231 (left) and HT1080 (right) cells treated with erastin at different concentrations for 12h, followed by immunoblots using a PKCβII-specific antibody to establish the interaction of endogenous AAK1 with endogenous PKCβII. Statistical analysis was performed using an unpaired two-tailed Student's t-test, * $p < 0.01$.

Extended Data Fig. 4G-J

Extended Data Fig. 4G-J. G, H, The co-localization of PKC β II with AAK1 performed by immunofluorescence in MDA-MB-231 (G) and HT1080 (H) cells treated with erastin at different concentrations for 12h. I, J, The co-localization of PKC β II with AAK1 performed by immunofluorescence in MDA-MB-231 (I) and HT1080 (J) cells treated with 10 μM or 5 μM erastin respectively for different time periods. Statistical analysis was performed using an unpaired two-tailed Student's t-test, * $p < 0.01$.

Extended Data Fig. 5G

Extended Data Fig. 5G. Purified AAK1 proteins were incubated with recombinant activated PKC β II kinase with/without λ -phosphatase for 0.5h in kinase buffer with ATP in vitro.

Fig. 3E, F, K

Fig. 3E, F, K. (E), Purified recombinant AAK1 proteins were incubated with/without recombinant activated PKCβII kinase or λ-phosphatase for 0.5h in kinase buffer with ATP in vitro. (F), HEK293T cells were transfected with Flag-AAK1 plasmid, followed by isolating from HEK293T cells using anti-Flag antibody. The isolated AAK1 proteins were incubated with/without recombinant activated PKCβII kinase or λ-phosphatase for 0.5h in kinase buffer with ATP in vitro. (K) HEK293T cells were transfected with wild-type Flag-AAK1 or mutant Flag-AAK1-S670/T674Ala plasmid, followed by isolating from HEK293T cells using anti-Flag antibody. The isolated AAK1 proteins were incubated with/without recombinant activated PKCβII kinase for 0.5h in kinase buffer with ATP in vitro. The phosphorylation levels of AAK1 were established by a phospho-S670/T674-AAK1-specific (p-AAK1 (S670/T674)) antibody.

Fig. 4D, E,

Fig. 4D, E. (D), The phosphorylation levels of AP2M1 in AAK1-knockout MDA-MB-231 (top)

and HT1080 (bottom) cells treated with erastin at different concentrations for 12h. (E), The phosphorylation levels of AP2M1 in MDA-MB-231 (top) and HT1080 (bottom) cells treated with erastin at different concentrations with/without 10 μ M SGC-AAK1-1 for 12h.

5. Why did the authors use siRNA and gRNAs to deplete cells of AAK1 in Figure 5?

Response: We confirmed the effect of AAK1 on ferroptosis using two cell models including AAK1 knockout by single guide RNAs (sgRNAs) and AAK1 knockdown by small interfering RNA (siRNAs). AAK1 in the former cell model was steadily and persistently depleted, while the latter cell model only lasted for 3-7 days after transfection. It was more convinced that similar results were indicated in two cell models.

6. In Figure 6C, is the difference in expression of AAK1 in primary tumor tissues versus normal tissues significant? The negative correlation in Extended Data 9A/B is weak, which should be mentioned in the text. Is there a difference in survival in patients with high and low expression of PKC β II or AP2M1 in the same dataset as used for Fig. 6A/B?

Response: The difference in expression of AAK1 in primary tumor tissues versus normal tissues in Fig. 6C is significant. We titled the P value in Fig. 6C in the revised manuscript. The correlation between AAK1 and gene set containing anti-ferroptosis genes was reanalyzed using other datasets, which was more significant (**Extended Data Fig. 9C, D**). In addition, survival analysis in breast cancer patients with high or low expression of PKC β II or AP2M1 was performed. Consistent with AAK1, high expression of PKC β II or AP2M1 was significantly associated with prolonged overall survival in breast cancer patients (**Extended Data Fig. 9A, B**).

Fig. 6C

Fig. 6C. Relative expression analysis of AAK1 in primary tumor tissues of breast cancer or normal tissues based on TCGA samples.

Extended Data Fig. 9A-D

Extended Data Fig. 9A-D. A, B, Kaplan–Meier analysis of overall survival related to the expression of PKC β II (A) and AP2M1 (B) for breast cancer patients. C, D, Correlation analysis between the expression of AAK1 and GOBP gene set containing anti-ferroptosis genes based on TCGA dataset (C) and ICGC-BC dataset (D).

7. At what point after inoculation with cells were the mice treated with IKE?

Response: IKE was dissolved in solvent containing 5% DMSO, 40% PEG300, 5% Tween 80 and 50% saline and then intraperitoneally injected to mice at a dose of 40mg/kg. Administration was performed in the fourth day after the injection when the tumors reached 50-100 mm³. These details were explained in the part of Methods.

Minor point:

1. The following statement starting in line 110 needs a citation: “We observed that the levels of ferritin were generally increased during ferroptosis as previously reported”.

The same applies to the reference to prior literature in line 123.

Response: Thank you for the helpful suggestion. We have supplemented a citation in

the mentioned lines (**Please refer to highlighted sentence at Page 4 in the revised manuscript**).

2. The model is nice, but additional details on the biochemical reactions would enhance it.

Response: Thank you for the helpful suggestion. The details on the biochemical reactions have been described in the legends of Figure3 and the above response. (**Please refer to highlighted sentence at Page 15 in the revised manuscript**)

Referee#2

(Remarks to the Author)

The authors investigate how cells regulate iron uptake during ferroptosis, an open point in the field. They identify a kinase that mediates transferrin receptor endocytosis through the activation of AAK1. They show that AAK1 phosphorylates AP2M1 which increases the endocytosis of iron and promotes ferroptosis in a technically sound and mechanistic way. However, there are significant concerns about novelty, as the AAK1-AP2M1 regulation of endocytosis is well described. In addition, translation and human modeling are absent, reducing our enthusiasm for this paper.

Response: Thank you for thoughtful comments. Regarding your concerns about the innovation of our research, we would like to make an explanation in more detail. As described in the manuscript, regulation of clathrin-mediated endocytosis by AAK1-AP2M1 has been reported. However, there are specific mechanisms through which endocytosis is precisely controlled in various physiological and pathological processes. Particularly, it is not clear how the endocytosis of TFR1 is regulated during ferroptosis. In this study, we found that the induction of ferroptosis in cancer cells typically leads to increased ferritin levels, which paradoxically contradicts the anticipated rise in ferritin degradation as previously reported. We further confirmed that the endocytosis of TFR1 was commonly elevated during ferroptosis, indicating extracellular iron uptake serves as a critical iron supply for the initiation of ferroptosis. Through the endocytosis assays, PKC β II, a previously reported sensor of lipid peroxidation, was identified as a key kinase mediating TFR1 endocytosis through phosphorylation and activation of AAK1 during the ferroptotic process, which subsequently phosphorylates AP2M1 and facilitates the recruitment of clathrin to mediate the endocytosis of TFR1. Therefore, our innovation appears in two main areas. Firstly, we confirm that the extracellular iron uptake through clathrin-mediated endocytosis of TFR1 significantly contributes to the initiation of ferroptosis. Secondly, we demonstrate the dual role of PKC β II - not only serves as a sensor of lipid peroxidation which detects and amplifies the accumulation of lipid peroxides, but also actively coordinates cellular iron acquisition. These findings establish a mechanistic

link between lipid peroxidation and iron dyshomeostasis, revealing their synergistic interplay in driving ferroptosis execution. In conclusion, our research plays a crucial role in understanding the initiation of ferroptosis.

In addition, we are dedicated to exploring the implications of this research for clinical practice. As ionizing radiation (IR) was reported to induce ferroptosis of tumors in radiotherapy, we performed supplementary experiments to confirm the effect of AAK1 to radiosensitivity *in vivo*. AAK1-knockout and negative control MDA-MB-231 cells were inoculated into nude mice, followed by treatment with either DMSO or Lipro-1, with or without IR. Expectedly, ferroptosis induced by IR was inhibited by Lipro-1. Knockout of AAK1 significantly inhibited ferroptosis and lipid peroxidation, thereby promoting tumor growth (**Extended Data Fig. 10A-E**). These results indicate that AAK1 enhances the radiosensitivity of tumors to ionizing radiation. Activation of PKC β II-AAK1-AP2M1 pathway may produce a synergistic effect combined with radiotherapy.

Extended Data Fig. 10A-E

Extended Data Fig. 10A-E. A-C, Tumor volume and weight of xenograft tumors formed by the indicated MDA-MB-231 cells treated with DMSO or Lipro-1, with or without IR. Lipro-1, 10 μ M. IR, 10Gy. D, Lipid-peroxidation measurement for tumor cells isolated from the indicated tumors. E, Representative immunohistochemical images of 4-HNE in the indicated tumor tissues. B, Data

are presented as means \pm SD, n = 3 biologically independent experiments. *p<0.01, two-way ANOVA test. **C, D**, Data are presented as means \pm SD, n = 6 independent samples. *p<0.01, unpaired two-tailed Student's t test.

Major:

1. In Figures 1A, 1D and 1E, the authors confirmed that erastin treatment induced FTH1, and intracellular iron. It could be due to either the increased uptake or impaired export of iron. Therefore, the expression of major iron transporters including TFR1, DMT1, FPN1 should be tested. Meanwhile, the recycling of TFR1 should be tested.

Response: Thank you for the helpful suggestion. In the manuscript, we observed that the iron levels were generally increased during erastin-induced ferroptosis in a panel of cancer cell lines (**Fig. 1B, C**). Based on recommendations, we detected the expression of major iron transporters including TFR1, DMT1, FPN1 in the revised manuscript. The results indicated that the expression of these proteins were not significantly changed, which support our previous conclusion that the increased iron levels during ferroptosis result from elevated endocytosis of TFR1 (**Fig. 1A**). In addition, we performed recycling assay of TFR1 with the treatment of primaquine, an inhibitor of endocytic recycling. PKC β -knockout cells were treated with 4 μ M erastin for 12h, followed by a time-course 37°C water bath in the addition of 200 μ M primaquine. As expected, primaquine induced an extra reduction of surface TFR1, suggesting that a proportion of internalized TFR1 was recycling (**Extended Data Fig. 1G**). Together, these results confirmed that the endocytosis of TFR1 was enhanced during ferroptosis in a PKC β -dependent manner, thereby increasing intracellular iron levels.

Fig. 1A-C

Fig. 1A-C. **A**, Levels of protein associated with iron metabolism were detected by immunoblotting in various cancer cell lines treated with erastin at different concentrations for 12h. A549, 8 μ M; CAL51, 30 μ M; HN6, 30 μ M; HT1080, 4 μ M; MDA-MB-231, 8 μ M; MCF7, 30 μ M. **B**, **C**, Total cellular iron levels (**B**) and divalent iron levels (**C**) were assayed in the indicated cancer cell lines treated with erastin at different concentrations for 12h. A549, 4 μ M; CAL51, 20 μ M; HN6, 20 μ M; HT1080, 1 μ M; MDA-MB-231, 2 μ M; MCF7, 20 μ M. Data are presented as means \pm SD, n = 3 biologically independent experiments. *p<0.01, unpaired two-tailed Student's t test.

Extended Data Fig. 1G

Extended Data Fig. 1G. Recycling assays of TFR1 in the indicated MDA-MB-231 cells treated with 4 μ M erastin for 12h, followed by treatment at 37 $^{\circ}$ C for different incubation time. Primaquine was treated with 200 μ M during incubation.

2. In Figure 1C, almost no endocytosis was detected in CAL51 and HN6 cells, indicating that there were some technical issues. Probably it was due to only 5-min incubation at 37 degrees. Extending the incubation time may help.

Response: Thank you for the helpful suggestion. According to our early exploration, incubation time of water bath significantly affects TFR1 endocytosis, which differs

from cell lines. We realized that the point mentioned above deserves attention. Therefore, we performed endocytosis assay under an improved condition including extension of incubation time from 5min to 10min in CAL51 and HN6 cells. In addition, we blocked endocytosis with dynasore, an endocytosis inhibitor which competitively blocks the GTPase activity of dynamin, as control group. New data reinforced the conclusion that the endocytosis of TFR1 was commonly elevated during ferroptosis (**Fig. 1E**).

Fig. 1E

Fig. 1E. Endocytosis assays of TFR1 were performed in various cancer cell lines treated with erastin at different concentrations for 12h. A549, 4 μ M; CAL51, 20 μ M; HN6, 20 μ M; HT1080, 1 μ M; MDA-MB-231, 2 μ M; MCF7, 20 μ M. Statistical analysis was performed using an unpaired two-tailed Student's t-test, * $p < 0.01$.

3. In Figure 2A, the authors did phosphoproteomic analysis. In this analysis, many targets were found, including AAK1. AAK1 was not the most dramatically changed gene. Therefore, the authors need to provide a better rationale for why AAK1 stands out.

Response: Thank you for raising a good point. As the manuscript reported, to explore the mechanism by which PKC β regulate the endocytosis of TFR1, we observed the phosphorylation levels of proteins in PKC β -knockout and negative control cells during ferroptosis through phosphoproteomic analysis (**Figure. 2A**). Actually, the phosphoproteomic analysis was not performed in the beginning of this study, but after we have confirmed the results that the endocytosis of TFR1 was commonly elevated during ferroptosis in a PKC β II dependent manner. Therefore, we performed the phosphoproteomic analysis based on two major reasons. Firstly, as PKC β II was

reported to be a significant kinase which regulates the phosphorylation of various proteins, we supposed that PKC β II may regulate TFR1 endocytosis through a similar mechanism. Secondly, to explore the mechanism of TFR1 endocytosis, proteins associated with endocytic process deserve more attention. Therefore, we considered that AAK1, a reported key kinase in the regulation of clathrin-mediated endocytosis, was a potential downstream protein through which PKC β II promotes the endocytosis of TFR1 during erastin-induced ferroptosis. This hypothesis was verified in subsequent experiments.

Fig. 2A

Fig. 2A. Phosphorylation level and gene score for individual genes were analyzed in the phosphoproteomic experiment using three individual wild type and PKC β -KO MDA-MB-231 cells. Potential genes exhibiting lower phosphorylation level have a negative score and genes exhibiting higher phosphorylation level have a positive score. Red highlights AAK1. Blue highlights ACSL4.

4. Figure 4A, the variation of GAPDH is too dramatic, which dampens the reliability of the data. For example, the increase of p-AP2M1 with erastin treatment could be caused by uneven protein loading.

Response: We have realized that this is a matter of concern. The newly provided data has addressed this issue (**Figure. 4A**).

Fig. 4A

Fig. 4A. The phosphorylation levels of AP2M1 in PKC β -knockout MDA-MB-231 (top) and HT1080 (bottom) cells treated with erastin at different concentrations for 12h.

5. Figures 4C and 4F, erastin treatment did not induce p-AP2M1 in NC cells, which is

controversial to previous data.

Response: Our results showed that the phosphorylation levels of AP2M1 was gradually elevated during ferroptotic process. The result in Figure 4C and 4F was performed using cell samples treated with erastin for the same time, which means ferroptosis may be observed only in sg1- PKC β II cells and sg1-S670/T674E cells but not in NC cells, because the former cells were more sensitive to ferroptosis. That's why AP2M1 was not phosphorylated in NC cells as previous data. We understand that this contradiction may cause concerns about the results. Therefore, we provided new data with improved condition by extending the treatment of erastin from 12h to 16h (Fig. 4C, F). Consistent with previous conclusion, these results showed elevated phosphorylation of AP2M1 during erastin-induced ferroptosis in NC cells.

Fig. 4C, F

Fig. 4C, F. (C), The phosphorylation levels of AP2M1 in the indicated MDA-MB-231 (top) and HT1080 (bottom) cells treated with/without 10 μ M or 5 μ M erastin respectively for 16h. (F), The phosphorylation levels of AP2M1 in the indicated MDA-MB-231 (top) and HT1080 (bottom) cells treated with/without 10 μ M or 5 μ M erastin respectively for 16h.

6. Figure 4F, WT AAK1 did not restore the phosphorylation of AP2M1.

Response: Consistent with the description above, the reason that WT AAK1 did not restore the phosphorylation of AP2M1 in Figure 4F may be the inappropriate treatment time of erastin. We have provided new data above, which has addressed this issue.

7. Figures 5I and 5K, only 20-30% cells died after erastin treatment, which is too low for a reliable conclusion.

Response: We performed these experiments with improved condition which extended the treatment of erastin from 12h to 16h (Fig. 5I, K). The ratio of dead cells was over 30%, which still supported our conclusion.

Fig. 5I, K

Fig. 5I, K. (I), Cell-death measurements for the indicated MDA-MB-231 (left) and HT1080 (right) cells treated with 10 μ M or 5 μ M erastin respectively for 14h. (K), Cell-death measurements for the indicated MDA-MB-231 (left) and HT1080 (right) cells treated with 10 μ M or 5 μ M erastin respectively for 14h. Statistical analysis was performed using an unpaired two-tailed Student's t-

test, *p<0.01.

8. Figure 6C, no significance was shown in the figure. If the data are not significant, the conclusion in rows 446-448 is not reasonable.

Response: This data was analyzed with TCGA database and directly exported. We ignored the remark of P value, which has been titled in the revised manuscript. Actually, P<0.001 (**Fig. 6C**).

Fig. 6C

Fig. 6C. Relative expression analysis of AAK1 in primary tumor tissues of breast cancer or normal tissues based on TCGA samples.

9. Figures 6E and 6F, the correlation analysis is too weak.

Response: To confirm the results, we increased the sample size from 162 to 217. These tissue chips were reprocessed and analyzed by HALO software (Indica Labs, version 3.6.4134). New data was more significant and still supported our previous conclusion (**Fig. 6E, F**).

Fig. 6E, F

E

PKCβII	p-AAK1		
	Low (%)	High (%)	Total
Low (%)	69 (31.8)	32 (14.7)	101 (46.5)
High (%)	37 (17.1)	79 (36.4)	116 (53.5)
Total	106 (48.9)	111 (51.1)	217

$\chi^2=28.661$ p < 0.01

F

PKCβII	4-HNE		
	Low (%)	High (%)	Total
Low (%)	52 (24.0)	26 (12.0)	78 (35.9)
High (%)	53 (24.4)	86 (39.6)	139 (64.1)
Total	105 (48.4)	112 (51.6)	217

$\chi^2=16.217$ p < 0.01

Fig. 6E, F. Correlation analysis between PKC β II and the phosphorylation level of AAK1 (**E**) or 4-HNE (**F**) based on tissue chips containing 217 human TNBC specimens. Statistical analysis was performed by Pearson Chi-square (χ^2) test.

Minor:

1. Figure 1D and 1E should be moved before endocytosis assay (1B and 1C), because 1D and 1E together with 1A demonstrated the increase of intracellular iron.

Response: We have already adjusted the position of these figures in the revised manuscript. **(Please refer to Figure 1 in the revised manuscript.)**

2. Extended Figure 1F, please indicate whether the cells were treated with erastin. If so, please specify the concentration and incubation time of the treatment.

Response: These cells were treated with 4 μ M (MDA-MB-231) or 2 μ M (HT1080) erastin for 12h respectively before 37 $^{\circ}$ C water bath. We have supplemented the information in the figure legends of the revised manuscript. **(Please refer to highlighted sentence at Page 40 in the legends of Extended Data Figure. 1F in the revised manuscript.)**

3. Row 230-233, PIK inhibitors decreased PKC β II-AAK1 interaction under erastin treatment. This does not suggest that PKC β II-AAK1 complex is involved in ferroptosis induction. The author over-interpreted the data.

Response: We realized the description was inappropriate. The sentence “the formation of the PKC β II-AAK1 complex was involved in the induction of ferroptosis” was revised to “the formation of the PKC β II-AAK1 complex was dependent on the activation of PKC β II” in the revised manuscript. **(Please refer to highlighted sentence at Page 10 in the revised manuscript.)**

4. Figures 2J and 2K are suggested to move to Figure 3. Figures 2J, 2K and 3 together confirms the phosphorylation of AAK1 by PKC β II.

Response: We have already adjusted the position of these figures in the revised manuscript. **(Please refer to Figure 2 and Figure 3 in the revised manuscript.)**

5. Row 282, the function of Go6983, enzastaurin needs to be explained.

Response: Go6983, enzastaurin were PKC inhibitors. We have explained it in the revised manuscript. **(Please refer to highlighted sentence at Page 12 in the revised manuscript.)**

6. Row 307-308, S670/T674 needs to be emphasized. Meanwhile, in Figures 3J and 3K, it is suggested to emphasize p-S670/T674 in the figures, even though this has been mentioned in the figure legends.

Response: We realized the phosphorylated sites of AAK1 should be emphasized. The issues mentioned above have been addressed in the revised manuscript. **(Please refer to highlighted sentence at Page 13 in the revised manuscript.)**

7. Row 420, data only suggest that PKC β II-AAK1-AP2M1 pathway affects ferroptosis sensitivity but not the induction of ferroptosis.

Response: We realized the description was inappropriate. The sentence “PKC β II-AAK1-AP2M1 pathway significantly contributes to the induction of ferroptosis” was revised to “PKC β II-AAK1-AP2M1 pathway significantly enhances the sensitivity to ferroptosis” in the revised manuscript. **(Please refer to highlighted sentence at Page 19 in the revised manuscript.)**

Referee#1

(Remarks to the Author)

The authors have mostly answered my questions. Their comment that the MS raw data will be uploaded soon after revision is not satisfactory; it needs to be completed and checked before final evaluation.

Response from authors:

Thank you for your thoughtful critiques and constructive comments that helped us to improve our manuscript. Please note that all data generated and analyzed in this study are included in the Article and its Supplementary Information files. The proteomics data has been submitted as a Supplementary Data File.